# Uncertainty for Active Learning on Graphs

## Abstract

Active learning (AL) is a promising technique to improve data efficiency of machine learning models by iteratively acquiring data labels during training. While Uncertainty Sampling (US)—a strategy that labels data points with the highest uncertainty—has proven effective for independent data, its implications for interdependent data, such as nodes in graphs, remain under-explored. In this work, we propose the first extensive study of US for node classification. Our contribution is threefold: **(1)** We are the first to provide a benchmark for US approaches beyond predictive uncertainty. We highlight a performance gap between conventional AL strategies for graphs and US. **(2)** We develop novel ground-truth Bayesian uncertainty estimates in terms of the data-generating process. We both theoretically prove and empirically confirm their effectiveness in guiding US toward high-quality label queries both on synthetic and real data. **(3)** Based on our analysis, we highlight pitfalls in modeling uncertainty related to contemporary estimators for node classification, enabling the development of principled US.

## 1 Introduction

Applications in machine learning are often limited by either the cost of acquiring new data labels like in experimental design (Sverchkov & Craven, 2017) or the cost of training on many labeled data (Cui et al., 2022). To remedy these problems, Active Learning (AL) allows the learner to query an oracle (e.g. users, machines, or experiments) to label specific data points considered *informative*, thus saving labeling labor and training effort that would have been spent on *uninformative* labeled data.

Uncertainty Sampling (US) methods (Beluch et al., 2018; Joshi et al., 2009) rely on uncertainty estimates to measure the informativeness of labeling each data point. Intuitively, areas where a learner lacks knowledge are assigned high uncertainty. Moreover, methods should distinguish the (irreducible) *aleatoric* uncertainty and the (reducible) *epistemic* uncertainty which the *total* uncertainty about a prediction is composed of (Kiureghian & Ditlevsen, 2009; Kendall & Gal, 2017). This disentanglement is particularly important for AL: The learner might not benefit much from labeling inherently uncertain instances while acquiring knowledge about data points for which uncertainty stems from reducible sources can be highly informative. This suggests epistemic uncertainty as sensible acquisition function (Nguyen et al., 2022).

For independent and identically distributed (i.i.d.) data, US methods for AL—in particular US methods disentangling aleatoric and epistemic uncertainty—have demonstrated high data efficiency and have been widely benchmarked (Kirsch et al., 2019; Beluch et al., 2018; Joshi et al., 2009; Gal et al., 2017; Nguyen et al., 2022). In contrast, in the case of interdependent data like graphs, e.g. for node classification, despite efforts to approach and benchmark AL (Zhu et al., 2003; Jun & Nowak, 2016; Regol et al., 2020; Wu et al., 2021; Zhang et al., 2022b), existing US methods neglect the multi-faceted nature of uncertainty and only facilitate total uncertainty (Cai et al., 2017; Gao et al., 2018; Li et al., 2022). This leaves it unclear (i) to which extent contemporary uncertainty estimators can effectively inform US for graph data, and (ii) whether disentangling aleatoric and epistemic uncertainty has similar benefits to AL as in i.i.d. settings. The complex nature of graph data makes investigating US methods for AL particularly challenging. Uncertainty estimates should not only capture information about node features independently, but also model information about their relationships (Stadler et al., 2021).

In this work, we delve into US for node classification problems. We critically evaluate and benchmark state-of-the-art uncertainty estimators against traditional AL strategies and find that US (as well as many other approaches) fall short of surpassing random sampling. Motivated by the effectiveness

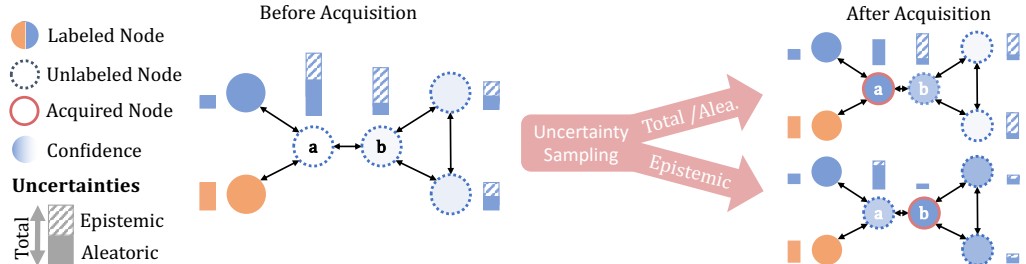

Figure 1: US can be realized maximizing either total, aleatoric or epistemic uncertainty. Former two include irreducible effects leading to label node $a$ while the latter isolates epistemic factors and queries the label of node $b$, thereby increasing the confidence in correctly predicting the remaining unlabeled nodes the most.

of US for i.i.d. data, we formally approach the question of whether US methods for graphs align well with the AL objective at all. We derive novel ground-truth uncertainty measures from the underlying data-generating process and disentangle its aleatoric and epistemic components. We prove that querying epistemically uncertain nodes is equivalent to maximizing the relative gain in probability a Bayesian classifier puts on the true labels of all other unlabeled nodes. Fig. 1 shows that acquiring the most epistemically uncertain label can improve the confidence of the classifier in the correct prediction more than selecting a node associated with high total or aleatoric uncertainty. By employing our proposed estimates on a Contextual Stochastic Block Model (CSBM) (Deshpande et al., 2018), we empirically confirm the validity of our findings in an idealized setting and generate valuable insights regarding principled uncertainty estimation in US: It is crucial to model uncertainty in terms of the true data generating process and *disentangle* it into aleatoric and epistemic components. Based on these results, we apply a simple approximation of disentangled ground-truth uncertainty to real data. Instead of proposing this as a novel AL strategy, we aim to show the applicability of our results to realistic scenarios. The main **contributions** of our work are:

- We provide the first extensive AL benchmark[1] for node classification including both a broad range of state-of-the-art uncertainty estimators for US and traditional acquisition strategies. We find many traditional methods and uncertainty estimators to not outperform random acquisition.

- We derive ground-truth aleatoric and epistemic uncertainty for a Bayesian classifier given the graph generative process and formally prove the alignment of US with AL.

- We empirically confirm the efficacy of epistemic US both using exact ground-truth uncertainty computed on a CSBM as well as a simple approximative method on real-world graphs that even outperforms other estimators on some datasets off-the-shelf.

## 2 BACKGROUND

**AL for Semi-Supervised Node classification.** Let $G = (\mathcal{V}, \mathcal{E})$, be a graph with a set of nodes $\mathcal{V}$ and edges $\mathcal{E} \subseteq \mathcal{V} \times \mathcal{V}$. With $n = |V|$ number of nodes the adjacency matrix is defined as $\boldsymbol{A} \in \{0, 1\}^{n \times n}$, where $A_{ij} = 1$ if there exists an edge between node $i$ and node $j$. The features are represented as matrix $\boldsymbol{X} \in \mathbb{R}^{n \times d}$. In node classification, every node has a corresponding label represented by a vector $\mathbf{y}$. We decompose the set of all nodes into $\mathcal{V} = \mathcal{U} \cup \mathcal{O}$ where $\mathcal{O}$ contains nodes with observed labels $\mathbf{y}_{\mathcal{O}}$ and $\mathcal{U}$ contains remaining unobserved nodes $\mathbf{y}_{\mathcal{U}}$ that we want to infer, where $\mathcal{V} = \mathcal{U} \cup \mathcal{O}$. We consider pool-based AL: In each iteration the learner queries an oracle for one label $\mathbf{y}_i$ from the set of unobserved labels $\mathbf{y}_{\mathcal{U}}$, adds it to the set of observed labels $\mathbf{y}_{\mathcal{O}}$ and retrains the model.

**Uncertainty in Machine Learning.** Alongside the predicted label $\mathbf{y}_i$, it is crucial to consider the associated uncertainty (Kiureghian & Ditlevsen, 2009; Kendall & Gal, 2017). Aleatoric uncertainty $u^{\text{alea}}$ is the inherent uncertainty that comes from elements like experimental randomness. It can not be reduced by acquiring more data. For instance, shooting at a target, external factors like wind can unpredictably affect the outcome. On the other hand, Epistemic uncertainty $u^{\text{epi}}$ reflects knowledge gaps, which can be addressed by data acquisition. In the previous example, this could mean adding shots at a new angle to the training data. A commonly employed conceptualization (Depeweg et al.,

---

[1]We provide our code at: `https://figshare.com/s/fd5e77eea5c8b71a7a7d`

2018) defines the total predictive uncertainty $u^{total}$ as encapsulating both reducible and irreducible factors, i.e. $u^{total} = u^{epi} + u^{alea}$.

**Contextual Stochastic Block Model.** We approach US on graphs sampled from an explicit generative process $p(\boldsymbol{A}, \boldsymbol{X}, \mathbf{y})$. To that end, we generate data from a Contextual Stochastic Blockmodel (CSBM) (Deshpande et al., 2018) enabling well-principled study of exact ground-truth uncertainty estimators. We first independently sample the node labels $\mathbf{y}$. Node features $\boldsymbol{X}$ and edges $\boldsymbol{A}$ are conditioned on these labels. We defer an in-depth description of this process to § C.3.

## 3 RELATED WORK

**Active Learning on Independent and Identically Distributed Data.** AL has seen substantial exploration in the context of i.i.d. data (Ren et al., 2021). Approaches can be divided into three main categories: diversity-based, uncertainty-based, or a combination thereof (Zhan et al., 2022). Diversity or representation-based methods query data samples that best represent the full dataset, i.e. they opt for a diverse set of data points. Approaches like KMeans or Coreset opt to minimize the difference in model loss between the selected training set and the whole data set (Sener & Savarese, 2018). Other approaches use adversarial techniques to estimate the representativeness and diversity of new samples (Sinha et al., 2019; Shui et al., 2020). Uncertainty-based approaches query instances that the classifier is most uncertain about. It is commonly computed from the predictive distribution of a classifier calculating its entropy (Shannon, 1948), the margin between the two most likely labels or as the probability of the least confident label (Wang & Shang, 2014). Houlsby et al. (2011) introduce Bayesian Active Learning by Disagreement (BALD) which queries points with high mutual information between the model parameters and the class label. Nguyen et al. (2019) leverage disentangled uncertainties (Kendall & Gal, 2017) for AL and propose that epistemic uncertainty is a better proxy for US than aleatoric estimates. Other approaches linearly combine diversity and uncertainty-based measures Yin et al. (2017) or employ a two-step optimization scheme Ash et al. (2020); Zhan et al. (2022).

**Active Learning on Interdependent Graph Data.** While a plethora of studies exist on AL for i.i.d. data, only a limited amount of work addresses interdependent data like graphs. Previous methods approach AL on graphs from different perspectives, including random fields (Zhu et al., 2003; Ma et al., 2013; Ji & Han, 2012; Berberidis & Giannakis, 2018), risk minimization (Jun & Nowak, 2016; Regol et al., 2020), adversarial learning (Li et al., 2020), knowledge transfer between graphs (Hu et al., 2020) or querying cheap soft labels (Zhang et al., 2022a). US is typically only considered in terms of the predictive distribution (Madhawa & Murata, 2020) which does not disentangle aleatoric and epistemic components. Also, other uncertainty-reliant approaches do not make that distinction (Cai et al., 2017; Gao et al., 2018; Li et al., 2022) even though the literature on i.i.d. data suggests that US benefits from disentangled uncertainty estimators (Nguyen et al., 2022; Sharma & Bilgic, 2016). The exploration of US for AL on graphs beyond total uncertainty remains largely uncharted territory. Our work targets this gap and showcases the unrealized potential of high-quality epistemic uncertainty estimators for AL on interdependent graph data.

**Uncertainty Estimation on Graphs.** US strategies necessitate accurate uncertainty estimates which can be obtained in different ways. In classification, the predictive distribution of deterministic classifiers like **GCN** (Kipf & Welling, 2017) and **APPNP** (Klicpera et al., 2018) has been used to obtain aleatoric uncertainty (Stadler et al., 2021). Bayesian approaches model the posterior distribution over the model parameters. **Ensembles** (Lakshminarayanan et al., 2017) fall into this category. They approximate the posterior over model parameters through a collection of independently trained models. **Monte Carlo Dropout** (MC-Dropout) Gal & Ghahramani (2016) instead emulates a distribution over model parameters by applying dropout at inference time. DropEdge (Rong et al., 2020) proposed to additionally drop edges to reduce over-fitting and over-smoothing. Variational Bayes methods place a prior on the model parameters (**BGCN**) and sample different parameter sets for each forward pass (Blundell et al., 2015). They allow access to both aleatoric and epistemic uncertainty estimates by approximating a distribution over predictions from multiple forward passes. Commonly employed measures of epistemic uncertainty are the mutual information between the model weights and predicted labels (Gawlikowski et al., 2023) or the variance in confidence about the predicted label (Stadler et al., 2021). Evidential methods like **GPN** (Stadler et al., 2021) disentangle epistemic and aleatoric uncertainty by outputting the parameters of a Dirichlet prior to the categorical

predictive distribution. This method has shown strong performance in detecting distribution shifts. Finally, Gaussian processes on graphs (Liu et al., 2020; Borovitskiy et al., 2021) have also proposed to model total uncertainty but do not disentangle aleatoric and epistemic uncertainty.

## 4 BENCHMARKING UNCERTAINTY SAMPLING APPROACHES FOR ACTIVE LEARNING ON GRAPHS

Previous studies on non-uncertainty-based AL on graphs find that AL strategies struggle to consistently outperform random sampling (Madhawa & Murata, 2020). We, therefore, ask the question of whether US shows any merit when using state-of-the-art uncertainty estimation. We are the first to design a comprehensive AL benchmark for node classification that not only encompasses traditional methods but also includes an extensive suite of contemporary uncertainty estimators for US.

> *Does Uncertainty Sampling using state-of-the-art uncertainty estimators work on graph data?*
> **No**, we find no uncertainty estimator to outperform random sampling. Further, most non-uncertainty strategies fail to consistently yield significant improvement over random queries.

**Experimental Setup.** We evaluate AL on five common citation benchmark datasets for node classification: **CoraML** (Bandyopadhyay et al., 2005), **Citeseer** (Sen et al., 2008; Giles et al., 1998), **PubMed** (Namata et al., 2012) as well as the co-purchase graphs **Amazon Photos** and **Amazon Computers** (McAuley et al., 2015). We evaluate the models of § 3: **GCN**, **APPNP**, **MC-Dropout**, **BGCN**, **GPN** and **Ensembles** and report average results over multiple AL runs (see § C).

**Baselines.** We benchmark all approaches against three baselines: (i) **Random.** Choose a node to label uniformly at random. (ii) **Balanced** Same as Random, but keeping the training balanced in terms of class labels. (iii) Optimized Final Accuracy **(OFA) Pool.** Selecting nodes from a predefined pool that was optimized to lead to strong classification performance. We provide an in-depth discussion about this baseline in § B. Note that apart from random acquisition, neither of these baselines can be employed for AL in practice as they rely on unavailable label information. We include them to highlight the existence of informative label sets.

Acquiring multiple labels per iteration can be challenging since greedily selecting the most promising candidates might overestimate the performance improvement (Kirsch et al., 2019). We therefore only acquire a single instance in each iteration, enabling us to analyze the performance of different acquisition strategies without having to consider potential side effects stemming from batched acquisition. We initially label one node per class and fix the acquisition budget to $4C$, a threshold at which the OFA pool achieves competitive classification performance.

In addition to a qualitative evaluation of how the accuracy of each classifier evolves when querying according to different strategies in Fig. 2, we report the accuracy of a classifier after the labeling budget is exhausted in Tab. 5. We expect good acquisition functions to achieve higher accuracy at a lower amount of queries, which in turn results in a larger area under (AUC) of the visualized curves. Normalizing the number of queries to $[0, 1]$, this metric corresponds average accuracy over the entire run, which we summarizes AL in Tab. 1.

We proceed as follows: First, we benchmark traditional AL approaches and find that only one outperforms random sampling. Then, we show that US fails to even match the performance of random queries in many instances regardless of its successful application to i.i.d. data. We supply the corresponding AUC and final accuracy scores as well as visualizations on all datasets in § D.

**Non-Uncertainty-based Strategies.** We first examine strategies that do not exclusively rely on uncertainty. (i) **Coreset** (Sener & Savarese, 2018) opts to find a core-set cover of the training pool by selecting nodes that maximize the minimal distance in the latent space of a classifier to already acquired instances. (ii) **Coreset-PPR** is similar to Coreset, but we use inverse PPR scores as a distance measure to select structurally dissimilar nodes. (iii) **Coreset Features.** Distances between nodes are computed only in terms of input features. (iv) **Degree** and **PPR.** We acquire nodes with the highest corresponding centrality measure. (v) **AGE** (Cai et al., 2017) and **ANRMAB** (Gao et al., 2018) combine total predictive uncertainty, informativeness, and representativeness metrics. (vi) **GEEM** (Regol et al., 2020) uses risk minimization to select the next query. Because of its high computational cost, we follow its authors and employ an SGC (Wu et al., 2019) backbone. (vii) **SEAL (Li et al., 2020) uses ad-**

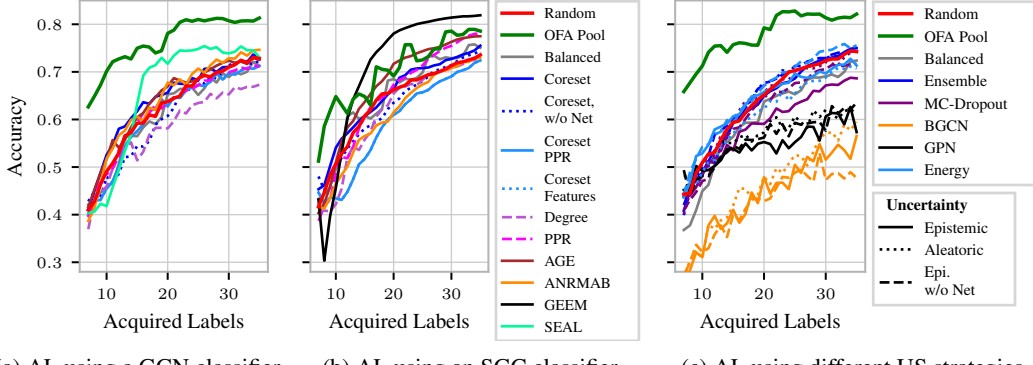

Figure 2: Accuracy of AL strategies on CoraML. Except for SEAL and GEEM, which is only tractable for SGCs, traditional AL and US can not match the OFA pool or outperform random queries.

versarial learning to identify nodes dissimilar from the labeled set. When applicable, we also consider setting $A = I$ during evaluation to exclude structural information similar to Stadler et al. (2021).

Observations: Figs. 2a and 2b show that only GEEM achieves performance similar to querying from the OFA pool on some datasets. The performance of GEEM comes at a high computational cost, as it requires training $\mathcal{O}(nC)$ models in each acquisition which makes it intractable for larger datasets and models beyond SGC. Structural Coreset strategies well on both co-purchase networks but do not show strong results on other graphs. This highlights a vast gap between many commonly used acquisition functions and a potential optimal strategy.

**Uncertainty Sampling.** We evaluate US using different uncertainty estimators: (i) **Aleatoric.** Similar to Stadler et al. (2021), we compute $u^{\text{alea}} = -\max_c \mathbf{p}_c$. (ii) **Epistemic.** For models that provide access to multiple predictions (MC-Dropout, Ensembles, BGCNs), we compute epistemic uncertainty as the variance of the confidence $u^{\text{epi}} = \text{Var}\left[\mathbf{p}_{\hat{c}}\right]$ in the predicted class $\hat{c}$. For GPN, we use evidence as a measure of epistemic confidence. (iii) **Energy.** Energy-based models (EBMs) (Liu et al., 2021; Wu et al., 2023) relate uncertainty to the energy $u = -\tau \log \sum_c \exp\left(\mathbf{l}_i\right)$ of the predicted logits $\mathbf{l}$. We apply this estimator to the deterministic GCN, APPNP, and SGC models as a surrogate for epistemic uncertainty. Again, we also consider corresponding variants of these strategies in which we keep structural information from the classifier.

| | | **Baselines** | | | **Non-Uncertainty** | | | | | | | | **Uncertainty** | | | |
|---|---|---|---|---|---|---|---|---|---|---|---|---|---|---|---|---|
| | **Inputs** | | | | | *A & X* | | | | *A* | *X* | *A & X* | | *X* | | |
| | Model | OFA Pool* | Random | Balanced* | Coreset | AGE | ANRMAB | GEEM | SEAL | Coreset PPR | Coreset Inputs | Epi./ (Energy) | Alea. | Epi./ (Energy) | Alea. |
| CoraML | GCN | **76.65** | 62.51 | 61.16 | 64.35 | 64.12 | 64.24 | n/a | 66.07 | 59.53 | 61.26 | 63.97 | 61.30 | 65.65 | 64.33 |
| | APPNP | **76.10** | 67.72 | 65.39 | 67.32 | 66.12 | 69.49 | n/a | n/a | 71.04 | 64.49 | 64.92 | 67.68 | 69.59 | 66.69 |
| | Ensemble | **77.98**† | 63.89 | 60.32 | 60.55 | 64.80 | 65.10 | n/a | n/a | 62.65 | 65.07 | 63.47 | 64.03 | 64.80 | 65.82 |
| | MC-Dropout | **76.89** | 64.94 | 62.31 | 64.37 | 64.44 | 64.06 | n/a | n/a | 62.92 | 64.35 | 59.17 | 63.69 | 61.82 | 63.87 |
| | BGCN | **58.42** | 45.76 | 45.98 | 49.37 | 51.25 | 47.23 | n/a | n/a | 39.43 | 44.85 | 44.45 | 46.61 | 42.74 | 48.11 |
| | GPN | **62.66** | 56.50 | 56.69 | n/a | n/a | n/a | n/a | n/a | 58.04 | 54.02 | 54.75 | 57.16 | 55.89 | 57.21 |
| | SGC | 70.48 | 63.85 | 65.04 | 65.23 | 67.56 | 61.14 | **71.39** | n/a | 60.24 | 59.18 | 67.51 | 65.66 | 65.05 | 67.13 |
| Pubmed | GCN | **70.34** | 61.56 | 63.58 | 62.61 | 69.48 | 60.31 | n/a | 58.62 | 61.71 | 56.71 | 59.64 | 61.85 | 59.66 | 60.34 |
| | APPNP | **75.09**† | 64.61 | 66.25 | 63.88 | 70.18 | 63.83 | n/a | n/a | 64.21 | 56.87 | 63.09 | 62.37 | 62.23 | 63.95 |
| | Ensemble | **71.05** | 59.26 | 61.30 | 64.25 | 68.26 | 60.40 | n/a | n/a | 61.89 | 56.36 | 63.70 | 61.37 | 59.71 | 61.15 |
| | MC-Dropout | **70.55** | 58.30 | 62.32 | 62.97 | 65.24 | 60.50 | n/a | n/a | 61.43 | 56.01 | 58.67 | 59.07 | 59.23 | 62.22 |
| | BGCN | **65.82** | 53.59 | 54.99 | 59.29 | 56.93 | 52.68 | n/a | n/a | 53.40 | 51.40 | 55.19 | 52.81 | 57.09 | 54.62 |
| | GPN | **73.35** | 56.79 | 61.37 | n/a | n/a | n/a | n/a | n/a | 62.08 | 54.34 | 58.82 | 57.24 | 56.25 | 59.37 |
| | SGC | **74.29** | 56.79 | 59.90 | 64.48 | 69.20 | 60.49 | 64.82 | n/a | 62.15 | 52.25 | 62.04 | 61.55 | 61.04 | 60.74 |
| AmazonPhotos | GCN | **84.37** | 79.06 | 78.72 | 78.58 | 75.17 | 79.97 | n/a | 71.16 | 70.20 | 82.71 | 74.66 | 74.61 | 79.96 | 79.63 |
| | APPNP | 81.30 | 79.29 | 80.65 | 81.04 | 79.02 | 80.35 | n/a | n/a | 76.37 | 84.24 | 79.72 | 77.45 | 80.48 | 77.69 |
| | Ensemble | **86.37** | 82.23 | 80.50 | 80.44 | 77.45 | 82.77 | n/a | n/a | 74.93 | 84.04 | 84.46 | 77.85 | 80.50 | 81.25 |
| | MC-Dropout | **84.34** | 80.32 | 76.39 | 76.63 | 74.75 | 80.21 | n/a | n/a | 75.32 | 82.45 | 72.42 | 73.16 | 69.68 | 78.80 |
| | BGCN | **75.24** | 71.22 | 70.19 | 67.15 | 65.69 | 70.69 | n/a | n/a | 59.34 | 73.39 | 70.83 | 67.83 | 72.21 | 69.19 |
| | GPN | 56.85 | 62.80 | 60.69 | n/a | n/a | n/a | n/a | n/a | 55.59 | **65.07** | 54.78 | 60.53 | 62.90 | 62.41 |
| | SGC | 84.46 | 80.52 | 79.24 | 82.32 | 74.01 | 80.92 | **86.43**† | n/a | 66.94 | 84.24 | 84.01 | 71.43 | 80.75 | 76.38 |

Table 1: Average AUC (↑) for different acquisition strategies on different models and datasets. We mark the best strategy per model in bold and underline the second best. For each dataset, we additionally mark the overall best model and strategy with the † symbol. Acquisitions that require access to class labels of non-train nodes are highlighted by an asterisk*.

Observations: Contrary to what the literature of US for i.i.d. data suggests (Beluch et al., 2018; Gal et al., 2017; Nguyen et al., 2022), we observe none of the US approaches to be effective in Fig. 2c.

While sampling instances with high aleatoric uncertainty matches the performance of random queries, we find epistemic uncertainty to even underperform in many instances. This is surprising, as the efficacy of epistemic US has been demonstrated for i.i.d. data (Nguyen et al., 2022; Kirsch et al., 2019). Only ensemble models match the performance of random sampling and slightly outperform on some datasets. GPN and energy-based approaches can not guide US toward effective queries. This is an intriguing result as both uncertainty estimators have shown to be highly effective when being assessed from the perspective of out-of-distribution detection (Stadler et al., 2021; Wu et al., 2023).

## 5 GROUND-TRUTH UNCERTAINTY FROM THE DATA GENERATING PROCESS

None of the established uncertainty-based methods yield satisfactory results. Consequentially, in this Section, we theoretically investigate whether US aligns with AL on graphs at all.

> *Can Uncertainty Sampling on graphs work effectively for AL?*
> **Yes**, we formally show that acquiring the node with maximal epistemic uncertainty optimizes the gain in the posterior probability of the ground-truth labels $\mathbf{y}_{\mathcal{U}}$ of unobserved nodes.

Evaluating the quality of uncertainty estimates is inherently difficult as generally, ground-truth values are unavailable for both the overall predictive uncertainty $u^{\text{total}}$ and its constituents $u^{\text{alea}}, u^{\text{epi}}$. Additionally, since epistemic uncertainty pertains to the knowledge of the classifier, it cannot be defined in a model-agnostic manner. To overcome these difficulties, we analyze uncertainty from the perspective of the underlying (potentially unknown) data-generating process $p(\boldsymbol{X}, \boldsymbol{A}, \mathbf{y})$ with respect to a Bayesian classifier. This view lends itself to a definition of ground-truth uncertainty. In the following, we propose confidence measures and relate them to uncertainty as their inverse $\text{conf} := \text{u}^{-1}$. This allows us to state the main theoretical result of our work: The optimality of US using epistemic uncertainty.

**Definition 1.** *We define the Bayesian classifier $f_\theta^*$, parameterized by $\theta$, in terms of the data generating process $p(\boldsymbol{A}, \boldsymbol{X}, \mathbf{y})$:*

$$f_\theta^*(\boldsymbol{A}, \boldsymbol{X}, \mathbf{y}_{\mathcal{O}}^{\text{gt}}) = \underset{\mathbf{c} \in \{1,\ldots,C\}^{|\mathcal{U}|}}{\operatorname{argmax}} \int \mathbb{P}\left[\mathbf{y}_{\mathcal{U}} = \mathbf{c} \mid \boldsymbol{A}, \boldsymbol{X}, \mathbf{y}_{\mathcal{O}} = \mathbf{y}_{\mathcal{O}}^{\text{gt}}, \theta\right] p(\theta \mid \boldsymbol{A}, \boldsymbol{X}, \mathbf{y}_{\mathcal{O}}) d\theta \quad (1)$$

Here, we denote with $\mathbf{y}_{\mathcal{O}}^{\text{gt}}$ the labels of already observed instances. The predictive distribution $p(\mathbf{y}_{\mathcal{U}} \mid \boldsymbol{A}, \boldsymbol{X}, \mathbf{y}_{\mathcal{O}}^{\text{gt}})$ encapsulates the total confidence of $f_\theta^*$. The classifier averages its prediction $\theta$ according to a learnable posterior distribution $p(\theta \mid \boldsymbol{A}, \boldsymbol{X}, \mathbf{y}_{\mathcal{O}})$ over its parameters. For example, $\theta$ could be parameters of a GNN fitting the data generating process. We define the total confidence associated with a single instance $\text{conf}^{\text{total}}$ as a marginal.

**Definition 2.** *The total confidence of $f_\theta^*$ in predicting label $c$ for node $i$ is defined as:*

$$\text{conf}^{\text{total}}(i, c) := \int \mathbb{P}\left[\mathbf{y}_i = c \mid \boldsymbol{A}, \boldsymbol{X}, \mathbf{y}_{\mathcal{O}} = \mathbf{y}_{\mathcal{O}}^{\text{gt}}, \theta\right] p(\theta \mid \boldsymbol{A}, \boldsymbol{X}, \mathbf{y}_{\mathcal{O}}) d\theta \quad (2)$$

Intuitively, the total confidence captures aleatoric factors through the inherent randomness of the joint distribution $p(\boldsymbol{A}, \boldsymbol{X}, \mathbf{y})$. Epistemic uncertainty is incorporated by conditioning on a limited set of observed labels $\mathbf{y}_{\mathcal{O}}^{\text{gt}}$. With a growing labeled set irreducible errors will increasingly dominate total predictive uncertainty. In the extreme case where all labels but one have been observed, i.e. $\mathcal{O} = \mathcal{V} \setminus \{v_i\}$, remaining uncertainty in the predictive distribution only stems from aleatoric factors.

**Definition 3.** *The aleatoric confidence of $f_\theta^*$ in predicting label $c$ for node $i$ is defined as:*

$$\text{conf}^{\text{alea}}(i, c) := \int \mathbb{P}\left[\mathbf{y}_i = c \mid \boldsymbol{A}, \boldsymbol{X}, \mathbf{y}_{-i} = \mathbf{y}_{-i}^{\text{gt}}, \hat{\theta}\right] p(\hat{\theta} \mid \boldsymbol{A}, \boldsymbol{X}, \mathbf{y}_{-i}) d\hat{\theta} \quad (3)$$

In this context, we denote with $\mathbf{y}_{-i} = \mathbf{y}_{-i}^{\text{gt}}$ that all nodes excluding the predicted node $i$ are observed as their true values. All remaining lack of confidence is deemed irreducible. Lastly, we define epistemic confidence by comparing aleatoric factors to the overall confidence. As both are defined probabilistically, it is natural to consider their ratio:

**Definition 4.** *The epistemic confidence of $f_\theta^*$ in predicting label $c$ for node $i$ is defined as:*

$$\text{conf}^{\text{epi}}(i, c) := \frac{\text{conf}^{\text{alea}}(i, c)}{\text{conf}^{\text{total}}(i, c)} \quad (4)$$

These definitions directly imply a notion of uncertainty: $u^{epi}(i,c) = u^{total}(i,c) / u^{alea}(i,c)$. Epistemic US thus labels node $i$ when the associated total uncertainty $u^{total}(i, \mathbf{y}_i^{gt})$ is large compared to its aleatoric uncertainty $u^{alea}(i, \mathbf{y}_i^{gt})$. Put differently, it favors uncertain nodes *when the uncertainty stems from non-aleatoric sources*. This naturally recovers the definition of epistemic uncertainty as the non-aleatoric component of the total predictive uncertainty. Since we can equivalently maximize the logarithm of the uncertainty, we also recover the well-established notion of the total uncertainty being a sum of aleatoric and epistemic factors:

$$\log u^{total}(i, \mathbf{y}_i^{gt}) - \log u^{alea}(i, \mathbf{y}_i^{gt}) = \log u^{epi}(i, \mathbf{y}_i^{gt}) \tag{5}$$

With these natural notions of confidence and uncertainty, we state the core result of our work:

**Theorem 1.** *A node that maximizes epistemic uncertainty at its true label $\mathbf{y}_i^{gt}$ maximizes the relative gain in terms of the posterior of the remaining true labels $\mathbf{y}_{\mathcal{U}-i}$. Hence, acquiring the most epistemically uncertain node is an optimal AL strategy for $f_\theta^*$.*

$$\operatorname{argmax}_i u^{epi}(i, \mathbf{y}_i^{gt}) = \operatorname{argmax}_i \frac{\mathbb{P}\left[\mathbf{y}_{\mathcal{U}-i} = \mathbf{y}_{\mathcal{U}-i}^{gt} \mid \boldsymbol{A}, \boldsymbol{X}, \mathbf{y}_{\mathcal{O}}, \mathbf{y}_i = \mathbf{y}_i^{gt}\right]}{\mathbb{P}\left[\mathbf{y}_{\mathcal{U}-i} = \mathbf{y}_{\mathcal{U}-i}^{gt} \mid \boldsymbol{A}, \boldsymbol{X}, \mathbf{y}_{\mathcal{O}}\right]} \tag{6}$$

We provide a proof of Thm. 1 in § A. Here, we refer to all unobserved labels excluding $\mathbf{y}_i$ as $\mathbf{y}_{\mathcal{U}-i}$. The ratio that is optimized by epistemic US corresponds to the relative increase in the posterior of the true unobserved labels. That is, it compares the probability the classifier $f_\theta^*$ assigns to the remaining unobserved true labels after acquiring the ground-truth label $\mathbf{y}_i^{gt}$ of node $i$ as opposed to not acquiring its label. High values indicate that the underlying classifier will be significantly more likely to predict the true labels of the remaining nodes after the corresponding query. Thus, a query that maximizes epistemic uncertainty will push the classifier toward predicting the true labels for all remaining unlabeled nodes. This holds for any Bayesian classifier that specifies a posterior $p(\theta \mid \boldsymbol{A}, \boldsymbol{X}, \mathbf{y}_{\mathcal{O}})$ over the parameters of the generative process. For example, fitting the parameters of a GNN can be interpreted as an instance of this framework. However, computing exact disentangled uncertainty requires access to unavailable labels $\mathbf{y}_{\mathcal{U}}$ and is therefore impractical. Hence, our analysis, motivates the development of tractable approximations to these quantities. Novel US approaches can directly benefit from the theoretical optimality guarantees that this work provides.

## 6 UNCERTAINTY SAMPLING WITH GROUND-TRUTH UNCERTAINTY

To support our theoretical claims, we employ US using the proposed ground-truth epistemic uncertainty as an acquisition function. Since for real-world datasets, the data generating process is not known, we first focus our analysis on CSBMs defined in § 2. This allows us to compute the uncertainty estimates of Defs. 2 to 4 directly by evaluating the explicit joint likelihood of the generative process $p(\boldsymbol{A}, \boldsymbol{X}, \mathbf{y})$ (see § E). While the optimality of epistemic US holds for any underlying data generating process, we focus on CSBMs as they have been extensively studied as surrogates for real data in node classification scenarios (Palowitch et al., 2022). To isolate the effect of correctly computing and disentangling uncertainty, we also assume the parameters of the underlying CSBM to be known to the Bayesian classifier $f_\theta^*$. Therefore, any discrepancies in US performance are purely linked to the suitability of different uncertainty estimation paradigms.

> *Can US be a strong acquisition strategy if it uses high-quality uncertainty estimates?*
> **Yes**, we observe a significant improvement over random acquisition when utilizing our proposed ground-truth uncertainty. We find it crucial for the uncertainty to **fully account for the generative process** and be **disentangled** in terms of aleatoric and epistemic factors.

We compare the performance of US using the proposed uncertainty measures to contemporary uncertainty estimators over 5 graphs with 100 nodes and 7 classes from sampled from a CSBM distribution $p(\boldsymbol{A}, \boldsymbol{X}, \mathbf{y})$. We report similar findings for larger graphs in § F.

Observations: Fig. 3 shows that, in general, the Bayesian classifier outperforms GNN architectures as it is aware of the true data generating process. In agreement with Thm. 1, we find epistemic uncertainty to not only significantly outperform random queries but also aleatoric and total uncertainty which we explain formally in the following. Props. 1 and 2. Further, epistemic US even outperforms

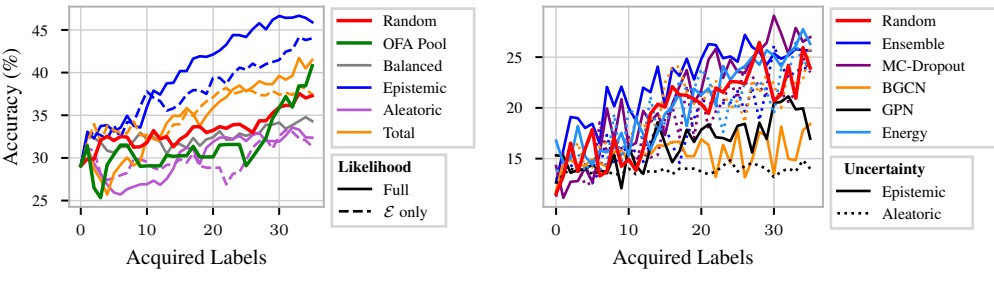

(a) AL using ground-truth uncertainty.    (b) AL using contemporary US strategies.

Figure 3: US on CSBM with 100 nodes and 7 classes. Ground-truth epistemic uncertainty significantly outperforms other estimators and random queries. Current US can not outperform random sampling.

the performance of the OFA pool already at a very small budget. At the same time, contemporary proxies for epistemic uncertainty do not consistently outperform random queries. Motivated by these observations, we analyze which aspects of uncertainty modelling are crucial to US and relate them to potential failure modes of current methods.

**Importance of Disentangling Uncertainty**. A natural question is why acquiring nodes with high total uncertainty $u^{\text{total}}$ performs worse than properly disentangling aleatoric and epistemic factors. We argue that total uncertainty favors not only informative queries but also tends to acquire labels of nodes that are associated with a high aleatoric uncertainty $u^{\text{alea}}$.

**Proposition 1.** *A node $i$ that maximizes total uncertainty maximizes the posterior probability of the remaining unobserved ground-truth labels $\mathbf{y}^{\text{gt}}_{\mathcal{U}-i}$ after acquiring its label $\mathbf{y}^{\text{gt}}_i$.*

$$\arg\max_i u^{\text{total}}(i, \mathbf{y}^{\text{gt}}_i) = \arg\max_i \mathbb{P}\left[\mathbf{y}_{\mathcal{U}-i} = \mathbf{y}^{\text{gt}}_{\mathcal{U}-i} \mid \mathbf{A}, \mathbf{X}, \mathbf{y}_{\mathcal{O}}, \mathbf{y}_i = \mathbf{y}^{\text{gt}}_i\right] \qquad (7)$$

We provide a proof of Prop. 1 in § A. Acquiring nodes with maximal total uncertainty maximizes the posterior of the remaining unlabeled set $\mathbf{y}_{\mathcal{U}-i}$. This is problematic as one way to increase this posterior probability is to remove an aleatoricly uncertain node $i$ from the unlabeled set. Such a query will not push the posterior of the remaining nodes in $\mathbf{y}_{\mathcal{U}}$ towards their true labels and instead improve the posterior by excluding nodes that are inherently difficult to predict. In contrast, the epistemic acquisition evaluates the joint posterior in relation to the effect of removing node $i$ from the unlabeled set (see Thm. 1). In fact, acquiring nodes with high aleatoric uncertainty directly opts to remove inherently ambiguous nodes from the unlabeled set.

**Proposition 2.** *A node $i$ that maximizes aleatoric uncertainty maximizes the posterior probability of the remaining unobserved ground-truth labels $\mathbf{y}^{\text{gt}}_{\mathcal{U}-i}$ without acquiring its label $\mathbf{y}^{\text{gt}}_i$.*

$$\arg\max_i u^{\text{alea}}(i, \mathbf{y}^{\text{gt}}_i) = \arg\max_i \mathbb{P}\left[\mathbf{y}_{\mathcal{U}-i} = \mathbf{y}^{\text{gt}}_{\mathcal{U}-i} \mid \mathbf{A}, \mathbf{X}, \mathbf{y}_{\mathcal{O}}\right] \qquad (8)$$

We provide a proof for Prop. 2 in § A. Prop. 2 explains why we observe aleatoric US to be ineffective in Fig. 3. It optimizes the posterior of the remaining labels $\mathbf{y}^{\text{gt}}_{\mathcal{U}-i}$ without considering the acquisition of $\mathbf{y}^{\text{gt}}_i$. Hence, such queries do not align with AL as they neglect the additional information obtained in each iteration. To optimize predictions on all remaining nodes it is crucial to properly disentangle uncertainty into aleatoric and epistemic components and acquire epistemically uncertain labels.

**Importance of Accurately Modelling the Generative Process**. We also highlight the importance of the uncertainty estimator to faithfully model the true data-generating process $p(\mathbf{A}, \mathbf{X}, \mathbf{y})$. To that end, we ablate our proposed uncertainty measures but only consider a Bayesian classifier that exclusively models present edges $(i, j) \in \mathcal{E}$ while neglecting $(i, j) \notin \mathcal{E}$. More precisely, we incorrectly model: $\hat{p}(\mathbf{A}, \mathbf{X}, \mathbf{y}) := \prod_{i<j,(i,j)\in\mathcal{E}} p(\mathbf{A}_{i,j} \mid \mathbf{y}_i, \mathbf{y}_j) \prod_i p(\mathbf{X}_i) \prod p(\mathbf{y}_i)$.

We specifically pick the inaccurate model to ignore non-existing edges because of its strong resemblance to contemporary GNN architectures used at the backbone of uncertainty estimators discussed in § 4. They rely on variations of the message-passing framework which propagates information exclusively along existing edges $\mathcal{E}$. In Fig. 3a, we observe that employing disentangled ground-truth uncertainty based on an inaccurate generative process neglecting non-existing edges harms US even when with proper uncertainty disentanglement. Thus, our analysis reveals another potential shortcoming of contemporary uncertainty estimators for graphs: They may fail to accurately learn the underlying data-generating process and thus be incapable of assessing uncertainty faithfully.

**Real-World Data.** Our theoretical analysis primarily focuses on the alignment of AL with epistemic US, employing ground-truth uncertainties for illustrative purposes which, in practice, are not available as they require full knowledge about the generative process and access to unavailable labels. We address this gap by proposing a simple approximate disentangled uncertainty estimator that facilitates an SGC classifier $f_\theta$. That is, we assume the classifier to model the unknown underlying generative process and interpret its predictive distribution as $\mathrm{conf}^{\mathrm{total}}$. We consider (i) computing the left-hand-side of Thm. 1 by approximating the log-difference between total and aleatoric uncertainty according to Defs. 2 and 3. The latter is computed from training a second classifier that utilizes pseudo-labels of $f_\theta$ for unobserved nodes. (ii) Approximating the numerator and denominator in the right-hand-side of Thm. 1 separately from the same pseudo-labels and training auxiliary classifiers. For details on the realization of both approaches, we refer to § H.

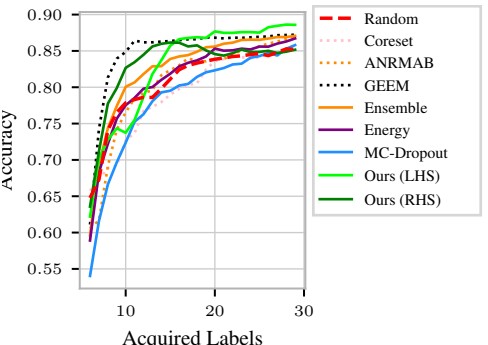

Figure 4: Approximation of our proposed uncertainty against top-performing traditional AL methods and US on Citeseer. Our framework works well either approximating the left-hand-side (LHS) or right-hand-side (RHS) of Thm. 1.

We visualize the performance of epistemic US using this approach in Fig. 4 and § H. On most datasets, this approach outperforms other US techniques. We also ablate a variant that directly optimizes the right-hand side of Thm. 1 instead (see § H). Note that both approaches introduce various sources of errors stemming from potentially incorrect pseudo-labels, miscalibration of $f_\theta$ or the classifier not faithfully modelling the data generating process, all of which our framework does not explicitly address. We do not mean to propose this approach as a novel AL strategy to be employed in practice. Instead, we underline that our theoretical analysis is highly applicable to practical applications as well since it significantly improves simple US strategies off-the-shelf.

**Pitfalls in US for Graphs.** Overall, we point out two crucial aspects in successfully employing US for graphs that may explain the lackluster performance of contemporary estimators: First, if the epistemic and aleatoric components are not *disentangled properly*, US misaligns with AL and favors uninformative queries. Second, if the model can not *accurately describe the underlying generative process*, AL performance also deteriorates significantly. Hence, uncertainty estimators need to isolate epistemic factors and should be designed with explicit consideration for the data generating-process.

## 7    LIMITATIONS

While our work reveals the potential of US for AL on graphs and the role of epistemic uncertainty as an optimal guide, we do not propose a concrete strategy but instead apply a straight-forward approximation of our theory on real-world data. This study serves as principled exploration into the landscape of US on graphs, aiming to inspire and inform future research in developing uncertainty estimators with consideration for the insights our study generates. In particular, while we find contemporary uncertainty estimators to underperform and show potential shortcomings, we leave the development of novel uncertainty estimation strategies that align well with AL to future work.

## 8    CONCLUSION

Our study sheds light on the potential and challenges of AL on graphs. An extensive benchmark reveals that most AL strategies only marginally outperform random queries at best, and contemporary uncertainty estimators inadequately guide US. We introduce ground-truth uncertainty estimates for node classification and formally demonstrate the alignment of US with AL. Our empirical analysis supports these insights on synthetic and real data, underscoring the necessity of accurately disentangling and modeling uncertainty estimates. We believe this to be a highly relevant result for uncertainty estimation on graphs that so-far neglected AL as an evaluation criterion. This work lays the necessary theoretical groundwork for developing well-principled uncertainty estimation techniques for interdependent data.

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

## A PROOFS

**Theorem 1.** *A node that maximizes epistemic uncertainty at its true label $\mathbf{y}_i^{\mathrm{gt}}$ maximizes the relative gain in terms of the posterior of the remaining true labels $\mathbf{y}_{\mathcal{U}-i}$. Hence, acquiring the most epistemically uncertain node is an optimal AL strategy for $f_\theta^*$.*

$$\operatorname{argmax}_i \mathrm{u}^{\mathrm{epi}}(i, \mathbf{y}_i^{\mathrm{gt}}) = \operatorname{argmax}_i \frac{\mathbb{P}\left[\mathbf{y}_{\mathcal{U}-i} = \mathbf{y}_{\mathcal{U}-i}^{\mathrm{gt}} \mid \boldsymbol{A}, \boldsymbol{X}, \mathbf{y}_\mathcal{O}, \mathbf{y}_i = \mathbf{y}_i^{\mathrm{gt}}\right]}{\mathbb{P}\left[\mathbf{y}_{\mathcal{U}-i} = \mathbf{y}_{\mathcal{U}-i}^{\mathrm{gt}} \mid \boldsymbol{A}, \boldsymbol{X}, \mathbf{y}_\mathcal{O}\right]}$$

*Proof.* We prove the stronger statement from which Thm. 1 follows directly.

$$\begin{aligned}
\mathrm{conf}^{\mathrm{alea}}(i, \mathbf{y}_i^{\mathrm{gt}}) &= \int \mathbb{P}\left[\mathbf{y}_i = \mathbf{y}_i^{\mathrm{gt}} \mid \boldsymbol{A}, \boldsymbol{X}, \mathbf{y}_{-i} = \mathbf{y}_{-i}^{\mathrm{gt}}, \theta\right] p(\theta \mid \boldsymbol{A}, \boldsymbol{X}, \mathbf{y}_{-i} = \mathbf{y}_{-i}^{\mathrm{gt}}) d\theta \\
&= \mathbb{P}\left[\mathbf{y}_i = \mathbf{y}_i^{\mathrm{gt}} \mid \boldsymbol{A}, \boldsymbol{X}, \mathbf{y}_{-i} = \mathbf{y}_{-i}^{\mathrm{gt}}\right] \\
&= \mathbb{P}\left[\mathbf{y}_i = \mathbf{y}_i^{\mathrm{gt}} \mid \boldsymbol{A}, \boldsymbol{X}, \mathbf{y}_\mathcal{O}, \mathbf{y}_{\mathcal{U}-i} = \mathbf{y}_{\mathcal{U}-i}^{\mathrm{gt}}\right] \\
&= \frac{\mathbb{P}\left[\mathbf{y}_{\mathcal{U}-i} = \mathbf{y}_{\mathcal{U}-i}^{\mathrm{gt}} \mid \boldsymbol{A}, \boldsymbol{X}, \mathbf{y}_\mathcal{O}, \mathbf{y}_i = \mathbf{y}_i^{\mathrm{gt}}\right]}{\mathbb{P}\left[\mathbf{y}_{\mathcal{U}-i} = \mathbf{y}_{\mathcal{U}-i}^{\mathrm{gt}} \mid \boldsymbol{A}, \boldsymbol{X}, \mathbf{y}_\mathcal{O}\right]} \mathbb{P}\left[\mathbf{y}_i = \mathbf{y}_i^{\mathrm{gt}} \mid \boldsymbol{X}, \boldsymbol{A}, \mathbf{y}_\mathcal{O}\right] \\
&= \frac{\mathbb{P}\left[\mathbf{y}_{\mathcal{U}-i} = \mathbf{y}_{\mathcal{U}-i}^{\mathrm{gt}} \mid \boldsymbol{A}, \boldsymbol{X}, \mathbf{y}_\mathcal{O}, \mathbf{y}_i = \mathbf{y}_i^{\mathrm{gt}}\right]}{\mathbb{P}\left[\mathbf{y}_{\mathcal{U}-i} = \mathbf{y}_{\mathcal{U}-i}^{\mathrm{gt}} \mid \boldsymbol{A}, \boldsymbol{X}, \mathbf{y}_\mathcal{O}\right]} \mathrm{conf}^{\mathrm{total}}(i, \mathbf{y}_i^{\mathrm{gt}}) \\
\mathrm{u}^{\mathrm{epi}}(i, \mathbf{y}_i^{\mathrm{gt}}) &= \frac{\mathrm{conf}^{\mathrm{alea}}(i, \mathbf{y}_i^{\mathrm{gt}})}{\mathrm{conf}^{\mathrm{total}}(i, \mathbf{y}_i^{\mathrm{gt}})} = \frac{\mathbb{P}\left[\mathbf{y}_{\mathcal{U}-i} = \mathbf{y}_{\mathcal{U}-i}^{\mathrm{gt}} \mid \boldsymbol{A}, \boldsymbol{X}, \mathbf{y}_\mathcal{O}, \mathbf{y}_i = \mathbf{y}_i^{\mathrm{gt}}\right]}{\mathbb{P}\left[\mathbf{y}_{\mathcal{U}-i} = \mathbf{y}_{\mathcal{U}-i}^{\mathrm{gt}} \mid \boldsymbol{A}, \boldsymbol{X}, \mathbf{y}_\mathcal{O}\right]}
\end{aligned}$$

First, we insert our definition of aleatoric confidence and marginalize $\theta$. Then, we split $\mathbf{y}_{-i}$ into two parts: observed $\mathbf{y}_\mathcal{O}$ and unobserved $\mathbf{y}_{\mathcal{U}-i}$. As we assign the ground-truth labels to both, the exact partition is not relevant. Next, we use Bayes law to get a distribution over the unobserved node labels $\mathbf{y}_\mathcal{U}$. Similarly to the first step, we marginalize $\theta$ to obtain $\mathrm{conf}^{\mathrm{total}}(i, \mathbf{y}_i^{\mathrm{gt}})$. In the last step, we see that the right term matches our definition of total confidence $\mathrm{conf}^{\mathrm{total}}$. $\square$

**Proposition 1.** *A node $i$ that maximizes total uncertainty maximizes the posterior probability of the remaining unobserved ground-truth labels $\mathbf{y}_{\mathcal{U}-i}^{\mathrm{gt}}$ after acquiring its label $\mathbf{y}_i^{\mathrm{gt}}$.*

$$\operatorname{argmax}_i \mathrm{u}^{\mathrm{total}}(i, \mathbf{y}_i^{\mathrm{gt}}) = \operatorname{argmax}_i \mathbb{P}\left[\mathbf{y}_{\mathcal{U}-i} = \mathbf{y}_{\mathcal{U}-i}^{\mathrm{gt}} \mid \boldsymbol{A}, \boldsymbol{X}, \mathbf{y}_\mathcal{O}, \mathbf{y}_i = \mathbf{y}_i^{\mathrm{gt}}\right]$$

*Proof.* We prove a stronger statement from which Prop. 1 follows directly.

$$\begin{aligned}
\mathrm{u}^{\mathrm{total}}(i, \mathbf{y}_i^{\mathrm{gt}}) &= \frac{1}{\mathrm{conf}^{\mathrm{total}}(i, \mathbf{y}_i^{\mathrm{gt}})} \\
&= \frac{\mathrm{conf}^{\mathrm{alea}}(i, \mathbf{y}_i^{\mathrm{gt}})}{\mathrm{conf}^{\mathrm{total}}(i, \mathbf{y}_i^{\mathrm{gt}})} \frac{1}{\mathrm{conf}^{\mathrm{alea}}(i, \mathbf{y}_i^{\mathrm{gt}})} \\
&= \frac{\mathbb{P}\left[\mathbf{y}_\mathcal{U} = \mathbf{y}_{\mathcal{U}-i}^{\mathrm{gt}} \mid \ldots, \mathbf{y}_i = \mathbf{y}_i^{\mathrm{gt}}\right]}{\mathbb{P}\left[\mathbf{y}_{\mathcal{U}-i} = \mathbf{y}_{\mathcal{U}-i}^{\mathrm{gt}} \mid \ldots\right]} \frac{1}{\mathrm{conf}^{\mathrm{alea}}(i, \mathbf{y}_i^{\mathrm{gt}})} \\
&= \frac{\mathbb{P}\left[\mathbf{y}_{\mathcal{U}-i} = \mathbf{y}_{\mathcal{U}-i}^{\mathrm{gt}} \mid \ldots, \mathbf{y}_i = \mathbf{y}_i^{\mathrm{gt}}\right]}{\mathbb{P}\left[\mathbf{y}_{\mathcal{U}-i} = \mathbf{y}_{\mathcal{U}-i}^{\mathrm{gt}} \mid \ldots\right]} \frac{1}{\mathbb{P}\left[\mathbf{y}_i = \mathbf{y}_i^{\mathrm{gt}} \mid \mathbf{y}_{\mathcal{U}_i} = \mathbf{y}_{\mathcal{U}-i}^{\mathrm{gt}}, \ldots\right]} \\
&= \frac{\mathbb{P}\left[\mathbf{y}_{\mathcal{U}-i} = \mathbf{y}_{\mathcal{U}-i}^{\mathrm{gt}} \mid \ldots, \mathbf{y}_i = \mathbf{y}_i^{\mathrm{gt}}\right]}{\mathbb{P}\left[\mathbf{y}_{\mathcal{U}-i} = \mathbf{y}_{\mathcal{U}-i}^{\mathrm{gt}} \mid \ldots\right]} \frac{\mathbb{P}\left[\mathbf{y}_{\mathcal{U}-i} = \mathbf{y}_{\mathcal{U}-i}^{\mathrm{gt}} \mid \ldots\right]}{\mathbb{P}\left[\mathbf{y}_i = \mathbf{y}_i^{\mathrm{gt}}, \mathbf{y}_{\mathcal{U}-i} = \mathbf{y}_{\mathcal{U}-i}^{\mathrm{gt}} \mid \ldots\right]} \\
&= \frac{\mathbb{P}\left[\mathbf{y}_{\mathcal{U}-i} = \mathbf{y}_{\mathcal{U}-i}^{\mathrm{gt}} \mid \ldots, \mathbf{y}_i = \mathbf{y}_i^{\mathrm{gt}}\right]}{\mathbb{P}\left[\mathbf{y}_i = \mathbf{y}_i^{\mathrm{gt}} \mathbf{y}_{\mathcal{U}-i} = \mathbf{y}_{\mathcal{U}-i}^{\mathrm{gt}} \mid \ldots\right]} \\
&= \frac{\mathbb{P}\left[\mathbf{y}_{\mathcal{U}-i} = \mathbf{y}_{\mathcal{U}-i}^{\mathrm{gt}} \mid \ldots, \mathbf{y}_i = \mathbf{y}_i^{\mathrm{gt}}\right]}{\mathbb{P}\left[\mathbf{y}_\mathcal{U} = \mathbf{y}_\mathcal{U}^{\mathrm{gt}} \mid \ldots\right]} \\
&\propto \mathbb{P}\left[\mathbf{y}_{\mathcal{U}-i} = \mathbf{y}_{\mathcal{U}-i}^{\mathrm{gt}} \mid \ldots, \mathbf{y}_i = \mathbf{y}_i^{\mathrm{gt}}\right]
\end{aligned}$$

Here, we abbreviated $\boldsymbol{A}, \boldsymbol{X}, \mathbf{y}_{\mathcal{O}}$ with $\ldots$ for clarity. First, we insert the result from Thm. 1. Then, we use the definition of conditional probability to replace the inverse of the aleatoric uncertainty. After canceling terms, we observe that $\mathbb{P}\left[\mathbf{y}_{\mathcal{U}} = \mathbf{y}_{\mathcal{U}}^{\mathrm{gt}} \mid \ldots\right]$ is a constant with respect to $i$ and arrive at the desired result that the total uncertainty is proportional to the posterior over the true labels $\mathbb{P}\left[\mathbf{y}_{\mathcal{U}-i} = \mathbf{y}_{\mathcal{U}-i}^{\mathrm{gt}}|\ldots, \mathbf{y}_i = \mathbf{y}_i^{\mathrm{gt}}\right]$. As in Prop. 1, we implicitly marginalized the learnable parameters $\theta$ of the Bayesian classifier.

□

**Proposition 2.** *A node $i$ that maximizes aleatoric uncertainty maximizes the posterior probability of the remaining unobserved ground-truth labels $\mathbf{y}_{\mathcal{U}-i}^{\mathrm{gt}}$ without acquiring its label $\mathbf{y}_i^{\mathrm{gt}}$.*

$$\operatorname{argmax}_i \mathrm{u}^{\mathrm{alea}}(i, \mathbf{y}_i^{\mathrm{gt}}) = \operatorname{argmax}_i \mathbb{P}\left[\mathbf{y}_{\mathcal{U}-i} = \mathbf{y}_{\mathcal{U}-i}^{\mathrm{gt}} \mid \boldsymbol{A}, \boldsymbol{X}, \mathbf{y}_{\mathcal{O}}\right]$$

*Proof.* We prove a stronger statement from which Prop. 2 follows directly.

$$
\begin{aligned}
\mathrm{u}^{\mathrm{alea}}(i, \mathbf{y}_i^{\mathrm{gt}}) &= \frac{1}{\mathrm{conf}^{\mathrm{alea}}(i, \mathbf{y}_i^{\mathrm{gt}})} \\
&= \frac{\mathrm{conf}^{\mathrm{total}}(i, \mathbf{y}_i^{\mathrm{gt}})}{\mathrm{conf}^{\mathrm{alea}}(i, \mathbf{y}_i^{\mathrm{gt}})} \frac{1}{\mathrm{conf}^{\mathrm{total}}(i, \mathbf{y}_i^{\mathrm{gt}})} \\
&= \frac{\mathbb{P}\left[\mathbf{y}_{\mathcal{U}-i} = \mathbf{y}_{\mathcal{U}-i}^{\mathrm{gt}}|\ldots\right]}{\mathbb{P}\left[\mathbf{y}_{\mathcal{U}} = \mathbf{y}_{\mathcal{U}-i}^{\mathrm{gt}} \mid \ldots, \mathbf{y}_i = \mathbf{y}_i^{\mathrm{gt}}\right]} \frac{1}{\mathrm{conf}^{\mathrm{total}}(i, \mathbf{y}_i^{\mathrm{gt}})} \\
&= \frac{\mathbb{P}\left[\mathbf{y}_{\mathcal{U}-i} = \mathbf{y}_{\mathcal{U}-i}^{\mathrm{gt}}|\ldots\right]}{\mathbb{P}\left[\mathbf{y}_{\mathcal{U}} = \mathbf{y}_{\mathcal{U}-i}^{\mathrm{gt}} \mid \ldots, \mathbf{y}_i = \mathbf{y}_i^{\mathrm{gt}}\right]} \frac{1}{\mathbb{P}\left[\mathbf{y}_i = \mathbf{y}_i^{\mathrm{gt}} \mid \ldots\right]} \\
&= \frac{\mathbb{P}\left[\mathbf{y}_{\mathcal{U}-i} = \mathbf{y}_{\mathcal{U}-i}^{\mathrm{gt}}|\ldots\right]}{\mathbb{P}\left[\mathbf{y}_{\mathcal{U}-i} = \mathbf{y}_{\mathcal{U}-i}^{\mathrm{gt}}, \mathbf{y}_i = \mathbf{y}_i^{\mathrm{gt}} \mid \ldots\right]} \\
&= \frac{\mathbb{P}\left[\mathbf{y}_{\mathcal{U}-i} = \mathbf{y}_{\mathcal{U}-i}^{\mathrm{gt}}|\ldots\right]}{\mathbb{P}\left[\mathbf{y}_{\mathcal{U}} = \mathbf{y}_{\mathcal{U}}^{\mathrm{gt}} \mid \ldots\right]} \\
&\propto \mathbb{P}\left[\mathbf{y}_{\mathcal{U}-i} = \mathbf{y}_{\mathcal{U}-i}^{\mathrm{gt}}|\ldots\right]
\end{aligned}
$$

Again, we abbreviated $\boldsymbol{A}, \boldsymbol{X}, \mathbf{y}_{\mathcal{O}}$ with $\ldots$ for clarity. First, we insert the result from Thm. 1 into $\mathrm{conf}^{\mathrm{total}}(i, \mathbf{y}_i^{\mathrm{gt}})/\mathrm{conf}^{\mathrm{alea}}(i, \mathbf{y}_i^{\mathrm{gt}})$. Then, insert the definition of total confidence. We apply the law of conditional probability, we see that $\mathbb{P}\left[\mathbf{y}_{\mathcal{U}} = \mathbf{y}_{\mathcal{U}}^{\mathrm{gt}} \mid \ldots\right]$ is a constant with respect to $i$ and arrive at the desired result that the aleatoric uncertainty is proportional to the posterior over the true labels $\mathbb{P}\left[\mathbf{y}_{\mathcal{U}-i} = \mathbf{y}_{\mathcal{U}-i}^{\mathrm{gt}}|\ldots\right]$.

□

## B   FINDING LABEL POOLS WITH STRONG PERFORMANCE

We benchmark AL strategies against a pool of labels that we empirically found to be performing well. That is, we trained GCN models with different 10,000 independent data splits and picked the training set that achieved the highest validation accuracy over five model initializations.

After fixing the set of queries, we optimize the order in which they are acquired by randomly sampling 2000 permutations of these labels for the citation networks CoraML, Citeseer, and PubMed and 500 random permutations for AmazonPhotos and AmazonComputers. Again, we select the ordering that yields the best AUC, accounting for a penalty term that discourages orderings in which the classification accuracy drops after acquiring new labels. Specifically, we pick the ordering $\mathcal{S}$ that optimizes:

$$\mathcal{L}(\mathcal{S}) = \frac{1}{5C} \sum_t \mathrm{acc}^{(t)}(\mathcal{S}) - 0.01 \sum_t \exp\left(\max(0, \mathrm{acc}^{(t-1)}(\mathcal{S}) - \mathrm{acc}^{(t)}(\mathcal{S}))\right) \tag{9}$$

While this OFA pool is optimized only on the performance of a GCN, we find it to provide an informative subset for most classifiers evaluated in § 4 on different datasets. Figs. 5a to 5e visualize average performances of AL runs using both the OFA pool for acquisition and random queries respectively. On most datasets, the OFA pool not only leads to a stronger performance of many classifiers after the budget is exhausted but also achieves higher accuracy with fewer queries. The exception is the AmazonComputers dataset, where the improvement in final accuracy over random acquisition is only marginal, however in lower data regimes the pre-selected pool exhibits some merit. In general, stronger classifiers are more capable of exploiting the informativeness provided by this pool while weaker models (BGCN, GPN) sometimes show inferior performance.

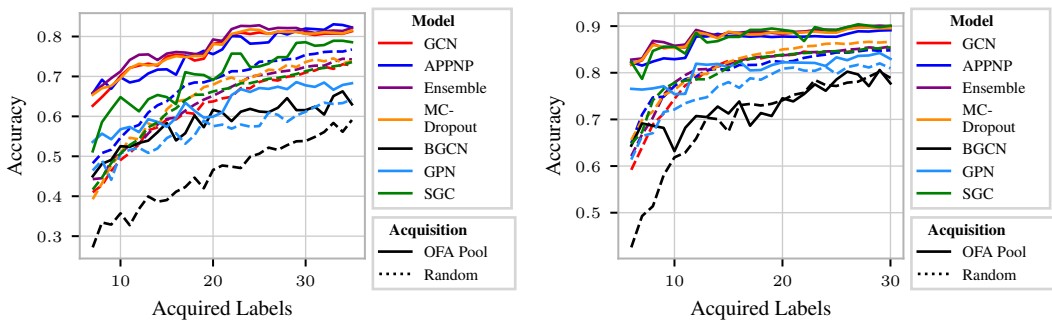

(a) Performance of different classifiers querying from the OFA pool versus random acquisition on CoraML.

(b) Performance of different classifiers querying from the OFA pool versus random acquisition on Citeseer.

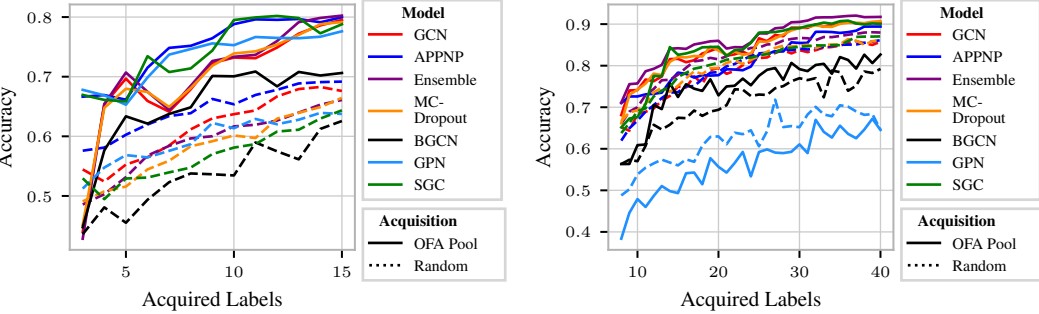

(c) Performance of different classifiers querying from the OFA pool versus random acquisition on PubMed.

(d) Performance of different classifiers querying from the OFA pool versus random on AmazonPhotos.

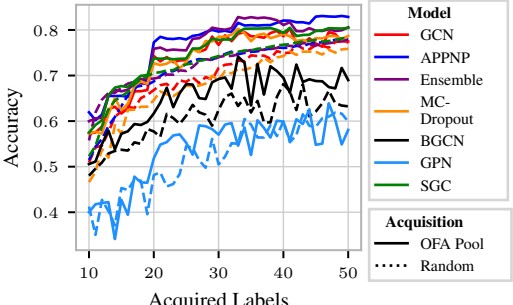

(e) Performance of different classifiers querying from the OFA pool versus random acquisition on Amazon-Computers.

Figure 5: Comparing the performance of acquiring nodes from the OFA pool optimized on a GCN on different classifiers. In most cases, this pool benefits models different than the GCN.

## C  EXPERIMENTAL SETUP

### C.1  ACTIVE LEARNING

We discuss AL on graphs, where given an initial set of labeled instances $\mathcal{O} \subset \mathcal{V}$ we aim to acquire a set of the unlabeled nodes $\mathcal{U} \subset \mathcal{V}$ that is optimal in terms of improving the performance of the classifier on the entire graph. That is, we (1) initially label one node randomly drawn from each class. (2) We then re-initialize the model weights and train the classifier until convergence. (3) We employ an acquisition strategy to select one or more unlabeled node(s). (4) We add the acquired label(s) to the training set and repeat the procedure at step (2) until some acquisition budget is exhausted. In contrast to domains where model training is expensive (Beluch et al., 2018; Gal et al., 2017; Kirsch et al., 2019), we re-train the classifier after each acquisition iteration. If not stated otherwise, we only acquire one node label in each iteration and fix the acquisition budget to $5C$. The resulting final training pools, therefore, contain fewer instances compared to dataset splits commonly used in other work (Kipf & Welling, 2017; Stadler et al., 2021).

### C.2  HYPERPARAMETERS

As hyperparameter tuning may be unrealistic in AL (Regol et al., 2020), we do not finetune them on a validation set. One potential strategy to realize hyperparameters is to randomly sample them from the search space over which hyperparameter optimization would be performed. This, however, is expected to lead to notably worse performance for most GNN architectures (Regol et al., 2020). Since the goal of our work is to showcase that even under optimal circumstances, contemporary uncertainty estimators can not enable US to be effective, we select one set of hyperparameters reported to be effective by the literature and employ it on all datasets. Specifically, we chose the following values:

| Model | Hidden Dimensions | Learning Rate | Max Epochs | Weight Decay | Teleport Probability | Power Iteration Steps | Dropout | Flow Dimension | Number Radial Flow Layers |
|---|---|---|---|---|---|---|---|---|---|
| GCN | [64] | 0.001 | 10,000 | 0.001 | n/a | n/a. | 0.0 | n/a | n/a. |
| APPNP | [64] | 0.001 | 10,000 | 0.001 | 0.2 | 10 | 0.0 | n/a | n/a |
| GPN | [64] | 0.001 | 10,000 | 0.001 | 0.2 | 10 | 0.5 | 16 | 10 |

Table 2: Hyperparameters of different GNN backbones

### C.3  DATASETS

**Real-World Datasets.** We evaluate AL approaches on three common node classification benchmark citation datasets: CoraML (McCallum et al., 2000; Sen et al., 2008; Bandyopadhyay et al., 2005; Giles et al., 1998), Citeseer (Sen et al., 2008; Bandyopadhyay et al., 2005; Giles et al., 1998), PubMed (Namata et al., 2012). In all datasets, nodes are papers and edges model citations. We consider undirected edges only and select the largest connected component in the dataset. All datasets use bag-of-words representations and we normalize the input features $\mathbf{x}_i$ node-wise to have a $L_2$-norm of 1. We report the statistics of each dataset in Tab. 3.

| Dataset | #Nodes $n$ | #Edges $m$ | #Input Features $d$ | #Classes $c$ | Edge Density $\frac{m}{n^2}$ | Homophily $p$ | Intra-Class Edge Density $p$ | Inter-Class Edge Density $q$ | SNR $\frac{p}{q}$ |
|---|---|---|---|---|---|---|---|---|---|
| CoraML | 2,810 | 15,962 | 2,879 | 7 | 0.20% | 78.44% | 1.69% | 0.06% | 28.51 |
| Citeseer | 1,681 | 5,804 | 602 | 6 | 0.20% | 92.76% | 4.89% | 0.03% | 141.36 |
| PubMed | 19,717 | 88,648 | 500 | 3 | 0.02% | 80.24% | 0.07% | 0.01% | 9.48 |
| AmazonComputers | 13,381 | 491,556 | 767 | 10 | 0.27% | 77.72% | 3.60% | 0.07% | 54.74% |
| AmazonPhotos | 7,484 | 238,086 | 745 | 8 | 42.47% | 82.72% | 3.77% | 0.10% | 36.85% |

Table 3: Dataset statistics.

We additionally compute the edge density $\frac{m}{n^2}$ as well as the homophily which is the fraction of edges that link between nodes of the same class. For comparability with CSBMs, we also report the average empirical inter-class edge probabilities $p$ and intra-class edge probabilities $q$ as well as their ratio, the structural signal-to-noise rate (SNR).

**CSBMs.**

CSBMs define the following generative process: First, node labels are sampled independently from a prior $p(\mathbf{y})$. For each node, features are then drawn independently from a class-conditional normal distribution $p(\mathbf{X}_i \mid \mathbf{y}_i) \sim \mathcal{N}(\mu_{\mathbf{y}_i}, \sigma_x^2 \mathbf{I})$. Each edge in the graph is generated independently according to an affiliation matrix $\mathbf{F} \in \mathbb{R}^{c \times c}$, i.e. $p(\mathbf{A}_{i,j} \mid \mathbf{y}_i, \mathbf{y}_j) \sim \text{Ber}(\mathbf{F}_{\mathbf{y}_i, \mathbf{y}_j})$. This gives rise to an explicit joint distribution over the graph that factorizes as $p(\mathbf{A}, \mathbf{X}, \mathbf{y}) = \prod_{i<j} p(\mathbf{A}_{i,j} \mid \mathbf{y}_i, \mathbf{y}_j) \prod_i p(\mathbf{X}_i \mid \mathbf{y}_i) \prod_i p(\mathbf{y}_i)$.

We generate CSBM graphs with a fixed number of nodes, classes, and input features according to § 2. That is, we first sample class labels $\mathbf{y}_i$ for each node independently from a uniform prior $\mathbb{P}[\mathbf{y}_i] = 1/C$. We then create links according to the affiliation matrix $\mathbf{F}$, where $\mathbb{P}[\mathbf{A}_{i,j} = 1 | \mathbf{y}_i, \mathbf{y}_j] = \mathbf{F}_{\mathbf{y}_i, \mathbf{y}_j}$. If not stated otherwise, we use a symmetric and homogeneous affiliation matrix $\mathbf{F}$, where we set all diagonal elements to a given intra-class edge probability $p$ and all off-diagonal elements to a given inter-class edge probability $q$. We enforce a structural signal-to-noise ratio (SNR) $\sigma_A = p/q$ by specifying an expected node degree $\mathbb{E}[\deg(v)]$ and then solving for:

$$q = \frac{\mathbb{E}[\deg(v)] c}{n-1} \frac{1}{\sigma_A + c - 1} \tag{10}$$

For each class, we first deterministically draw $C$ vectors of dimension $d$ that all have a pairwise distance of $\delta_X$. We then use a random rotation to obtain class centers $\mu_c$. For each node, we then independently sample its features $\mathbf{x}_i$ from a normal distribution $\mathcal{N}(\mathbf{X}_i | \mu_{\mathbf{y}_i}, \sigma_X \mathbf{I})$. We refer to the quotient $\frac{\delta_X}{\sigma_X}$ as the feature signal-to-noise ratio. The dimension of the feature space is given by $d = \max(c, \lceil n/\log^2 n \rceil)$, following Gosch et al. (2023).

## C.4 MODEL DETAILS

If applicable, we use the same GCN (Kipf & Welling, 2017) backbone for all models. That is, we employ one hidden layer of dimension of $64$. The APPNP (Klicpera et al., 2018) model uses an MLP with one hidden layer of the same dimension. We diffuse the predictions using 10 steps of power iteration and a teleport probability of $0.2$.

If not stated otherwise, ensembles are composed of 10 architecturally identical GCNs. In the case of MCD, we apply dropout with probability $0.1$ to each neuron of the GCN backbone independently. The BGCN mimics the GCN but models weights and biases as normal distributions. We follow (Blundell et al., 2015) and regularize these towards a standard normal distribution. Consequentially, we apply a weight of $\lambda = 0.1$ to the regularization loss.

We follow Stadler et al. (2021) to configure the GPN model: We use 10 radial flow layers to implement the class-conditional density model with a dimension of 16. For diffusion, we implement the same configuration as for APPNP. Other hyperparameters are set as in (Stadler et al., 2021).

For SEAL, we follow the authors and do not re-train the model after each acquisition (Li et al., 2020). Furthermore, we pick the number of training iterations for the discriminator to be $n_d = 5$ and the number of iterations for the generator $n_g = 10$, a combination that we observed to be successful on one dataset and kept fixed for all others as the authors do not provide them directly and hyperparameter optimization is unrealistic in AL (Regol et al., 2020).

## C.5 TRAINING DETAILS

Apart from the GPN, all of the aforementioned models are trained towards the binary-cross-entropy objective using the ADAM (Kingma & Ba, 2014) optimizer with a learning rate of $10^{-3}$ and weight decay of $10^{-3}$. We also perform early stopping on the validation loss with a patience of 100 iterations. We implement our models in PyTorch and PyTorch Geometric and train on two types of machines: (i) Xeon E5-2630 v4 CPU @ 2.20GHz with a NVIDA GTX 1080TI GPU and 128 GB of RAM. (ii) AMD EPYC 7543 CPU @ 2.80GHz with a NVIDA A100 GPU and 128 GB of RAM .

For each dataset, classifier, and acquisition function, we report results averaged over five different dataset splits and five distinct model initializations. In each dataset split, we a priori fix 20% of all nodes as a test set that is reused in every subsequent dataset split and can never be acquired by any strategy.

# D    ADDITIONAL METRICS AND PLOTS

We supply an evaluation of contemporary non-uncertainty-based acquisition strategies on different models as well as various uncertainty estimators for US over all datasets listed in § C.3. We briefly summarize the key insights:

(i) **CoraML.** Figs. 2a, 2c, 6a and 6b show that apart from GEEM no AL strategy is significantly more effective than random sampling. For an SGC classifier, GEEM outperforms the empirically determined OFA pool which was found on a GCN. While this pool appears to not perfectly translate to the SGC classifier for the CoraML dataset, GEEM can provide high-quality queries. In terms of US approaches, only ensemble methods perform somewhat better than random acquisition. (ii) **Citeseer.** Figs. 6c to 6f show that no AL strategy can match the performance of the OFA pool. Again, GEEM is the strongest performing approach even though it also can not identify a training set on par with the OFA pool. Only ensembles and energy-based models can compete with random acquisition when concerning epistemic uncertainty estimates. We verify the intriguing trend observed in § 4 that epistemic uncertainty proxies seem to significantly underperform random queries when being employed for US. This supports our conjecture that contemporary estimators may not properly disentangle or only partially model uncertainty. (iii) **PubMed.** In Figs. 6g to 6j, GEEM is outperformed by the AGE baseline. While it, too, can identify training sets that surpass random acquisition in terms of accuracy, it does not match the performance of the OFA pool. For this dataset, acquiring central nodes (PPR, AGE) appears to be a successful strategy. Interestingly, ANRMAB fails to exploit centrality despite it, in theory, being capable of doing so. US appears to be effective only when employing ensemble methods. (iv) **AmazonPhotos.** On this dataset, GEEM outperforms even the empirical OFA pool which is also matched by a centrality-based (PPR) Coreset approach (see Figs. 6k to 6n. Concerning US, we again find that only ensemble methods outperform random sampling while other estimators lead to significantly worse performance. (v) **Amazon Computers.** Similar to the other co-purchase network, we observe Coreset PPR to be a strong proxy for AL on the AmazonComputers dataset (Figs. 6o to 6r). While GEEM (which is only applicable to an SGC classifier) outperforms the OFA pool when compared to other methods on GCNs and APPNP, an SGC model trained on the OFA pool achieves better accuracy than GEEM. For this dataset, all US approaches–including ensembles–perform significantly worse than a random strategy.

This affirms the statements of § 4: Among non-uncertainty strategies, only GEEM can consistently outperform random and in some cases even match the performance of a strong training set. In turn, most US approaches consistently underperform random queries, a trend that may be indicative of improperly disentangling uncertainty. Only ensembles show small merit in most cases and, at least, do not fall short of random queries.

We supplement our findings by reporting the accuracy of each classifier after the acquisition budget is exhausted in Tab. 5. The corresponding rankings align well with the respective AUC scores Tab. 4. Both metrics are averaged over all 5 dataset splits and model initializations and we also report the standard deviations. By § 4, we conclude that all acquisition strategies fail to match the performance of the empirically determined OFA split.

# E  COMPUTING ALEATORIC AND TOTAL CONFIDENCE ON CSBMS

Computing the aleatoric confidence is straightforward directly employing Def. 3 using Bayes rule and the generative process up to a tractable normalization constant.

$$\text{conf}^{\text{alea}}(i,c) \propto p(\boldsymbol{A}, \boldsymbol{X} \mid \mathbf{y}_{-i}^{\text{gt}}, \mathbf{y}_i = c) p(\mathbf{y}_i = c) \tag{11}$$

Computing the epistemic confidence, however, turns out to have exponential complexity in the number of unlabeled nodes $|\mathcal{U} - i|$:

$$\text{conf}^{\text{epi}}(i,c) = p(\mathbf{y}_i = c \mid \boldsymbol{A}, \boldsymbol{X}, \mathbf{y}_{\mathcal{O}}^{\text{gt}}) \tag{12}$$

$$= \sum_{\mathbf{y}_{\mathcal{U}-i}} p(\mathbf{y}_i = c = \mathbf{y}_{\mathcal{U}-1} \mid \boldsymbol{A}, \boldsymbol{X}, \mathbf{y}_{\mathcal{O}}^{\text{gt}}) \tag{13}$$

$$\propto \sum_{\mathbf{y}_{\mathcal{U}-i}} p(\boldsymbol{A}, \boldsymbol{X} \mid \mathbf{y}_i = c, \mathbf{y}_{\mathcal{U}-i}, \mathbf{y}_{\mathcal{O}}^{\text{gt}}) p(\mathbf{y}_i = c, \mathbf{y}_{\mathcal{U}-i}) \tag{14}$$

**Approximating Epistemic Confidence**. For larger graphs, this quickly becomes intractable and we therefore rely on a variational mean-field approximation of the joint distribution $p(\mathbf{y}_{\mathcal{U}} \mid \boldsymbol{A}, \boldsymbol{X}, \mathbf{y}_{\mathcal{O}})$ to obtain marginals similar to Mariadassou et al. (2010); Jaakkola (2000).

$$p(\mathbf{y}_{\mathcal{U}} \mid \boldsymbol{A}, \boldsymbol{X}, \mathbf{y}_{\mathcal{O}}) \approx q(\mathbf{y}_{\mathcal{U}}) := \prod_{i \in \mathcal{U}} q_i(\mathbf{y}_i) \tag{15}$$

Since the variational distributions $q_i$ are discrete, we can fully describe them with parameters $\gamma_{i,c} := q_i(\mathbf{y}_i = c)$. The ELBO of this variational problem is given by:

$$J(\gamma) = \log p(\boldsymbol{X}, \boldsymbol{A}, \mathbf{y}_{\mathcal{O}}) + \mathbb{KL}\left[q(\mathbf{y}_{\mathcal{U}}) \,\middle\|\, p(\mathbf{y}_{\mathcal{U}} \mid \boldsymbol{A}, \boldsymbol{X}, \mathbf{y}_{\mathcal{O}})\right] \tag{16}$$

$$= \mathbb{E}_{\mathbf{y}_{\mathcal{U}} \sim q} \left[\log p(\boldsymbol{A}, \boldsymbol{X}, \mathbf{y}_{\mathcal{U}} \mid \mathbf{y}_{\mathcal{O}})\right] + \mathbb{H}\left[q(\mathbf{y}_{\mathcal{U}})\right] + \text{const} \tag{17}$$

$$= \sum_{i,c} \gamma_{i,c} \log p(\mathbf{y}_i = c) + \sum_{i<j} \sum_{c,c'} \gamma_{i,c} \gamma_{j,c'} \log p(\boldsymbol{A}_{i,j} \mid \mathbf{y}_i = c, \mathbf{y}_j = c') \tag{18}$$

$$+ \sum_{i,c} \gamma_{i,c} \log p(\boldsymbol{X}_i \mid \mathbf{y}_i = c) + \text{const} \tag{19}$$

Here, we introduced $\gamma_{j,c} = \mathbb{1}(c = \mathbf{y}_j^{\text{gt}})$ for all $j \in \mathcal{O}$ for convenience. This gives rise to a constrained optimization problem where $\sum_c \gamma_{i,c} = 1$ for all $i \in \mathcal{U}$. Solving this problem analytically gives rise to the equations:

$$\gamma_{i,c} \propto \exp\left(\log p(\mathbf{y}_i = c) + \sum_{j \neq i} \sum_{c'} \gamma_{j,c'} \log p(\boldsymbol{A}_{i,j} \mid \mathbf{y}_i = c, \mathbf{y}_j = c') + \log p(\boldsymbol{X}_i \mid \mathbf{y}_i = c)\right) \tag{20}$$

The marginal probabilities $\gamma_{i,c}$ can then directly be optimized by the resulting fixed point iteration scheme, where after each iteration the probabilities are normalized:

$$\log \gamma_{i,c}^{(t+1)} = \log p(\mathbf{y}_i = c) + \sum_{j \neq i} \sum_{c'} \gamma_{j,c'}^{(t)} \log p(\boldsymbol{A}_{i,j} \mid \mathbf{y}_i = c, \mathbf{y}_j = c') + \log p(\boldsymbol{X}_i \mid \mathbf{y}_i = c) \tag{21}$$

The quality of the acquisition function given by the epistemic confidence relies on the quality of the approximated marginals $p(\mathbf{y}_i \mid \boldsymbol{A}, \boldsymbol{X}, \mathbf{y}_{\mathcal{O}})$, i.e. the total confidence $\text{conf}^{\text{total}}(i, \cdot)$. To verify

that the proposed variational scheme indeed provides reasonable approximations, we report the absolute approximation errors $|q(\mathbf{y}_i = c) - p(\mathbf{y}_i = c \mid \boldsymbol{A}, \boldsymbol{X}, \mathbf{y}_{\mathcal{O}})|$ for CSBM graphs such that exact computation of the true marginals is tractable, i.e. up to 12 nodes.

Fig. 7 shows the distribution of approximation errors averaged over five different samples from a CSBM generative distribution with a structural SNR of $\sigma_A = 2.0$ and a feature SNR $\sigma_X = 1.0$ as well as an expected node degree $\mathbb{E}\left[\deg(v)\right] = 4.0$. In general, we expect the approximation to be of higher quality the more decoupled the marginals $p(\mathbf{y}_i \mid \boldsymbol{A}, \boldsymbol{X}, \mathbf{y}_{\mathcal{O}})$ are (Jaakkola, 2000). While the median error does not exceed $5\%$, we observe some outliers for larger graphs. In such cases, the employed approximation is inaccurate. Nonetheless, in Fig. 3 we observe that even in the face of sometimes poor approximations, the proposed uncertainty framework achieves strong results. We suspect a stronger approximation to perform even better.

## F    Uncertainty Sampling with Ground-Truth Uncertainty

In Fig. 8, we supplement our findings from Fig. 3 for graphs with 1000 nodes and 4 classes sampled from the same CSBM distribution. Again, we find epistemic US to be the strongest approach in terms of the Bayesian classifier while total uncertainty can not match its performance but also outperforms random sampling. We also observe a decrease in performance when the generative process is modelled incorrectly. Also contemporary estimators fail to outperform random queries, confirming the results reported in § 5 for larger graph sizes. We do not benchmark the Bayesian classifier against an OFA pool since approximation of the classifier marginals according to Eq. (21) becomes computationally taxing with an increasing graph size.

### F.1    Ablation of the Proposed Acquisition Strategy on Different CSBM Configurations

We also ablate the proposed acquisition strategy on different configurations of the CSBM. That is, we sample graphs from a CSBM with homogeneous and symmetric affiliation matrices and vary both the structural SNR $\frac{p}{q}$ as well as the feature SNR $\frac{\delta_X}{\sigma_X}$. The underlying CSBM has 100 nodes and 4 classes and labels are sampled from a uniform prior (see § C.3).

We perform AL on five graphs sampled from each configuration independently and measure the absolute improvement Uncertainty Sampling achieves in terms of AUC and accuracy after a budget of $5C = 20$ is exhausted. In Fig. 11a, we show the average improvement in AUC for each SNR configuration. We also supply a similar visualization for the accuracy after the budget is exhausted in Fig. 11b.

We find that our approach achieves the strongest improvement in both metrics when the structural SNR $\sigma_A$ is neither too high nor too small: Large values make the classification problem too easy and hence AL strategies do not have to carefully query informative nodes as the overall performance is strong even in very label scare regimes. Interestingly, our ground-truth uncertainty estimator shows strong merit when the node features are noisy, indicating that it is crucial to pick structurally informative nodes in the graph in these regimes.

At the same time, when the structural SNR is too low (in particular drops below $1.0$), our method fails to outperform random acquisition: We attribute this to the mean-field approximation to the total confidence $\mathrm{conf}^{\mathrm{total}}$ described in § E: We observed that for graphs with low structural SNR, the fixed point iteration of Eq. (21) does not converge. Hence, the approximated marginal probabilities are poor and both the prediction as well as the uncertainty estimation based on it deteriorate.

## G    Visualization of Ground-Truth Uncertainty on a Toy Example

In the following, we illustrate the behavior of this acquisition function on a small CSBM graph. Figure 12a shows a sample graph with three classes and nine nodes. The greyed-out histograms represent the aleatoric confidence. The distributions correspond to the total confidence, and the size of the nodes indicates the inverse of the ratio of both, i.e., the epistemic uncertainty, which guides our acquisition. We note that nodes 7 and 8 are the most promising candidates, as their aleatoric

prediction is confident, and in the initial step since all nodes are unlabeled, the total confidence is uniform across all nodes due to the homogenous structure of the affiliation matrix.

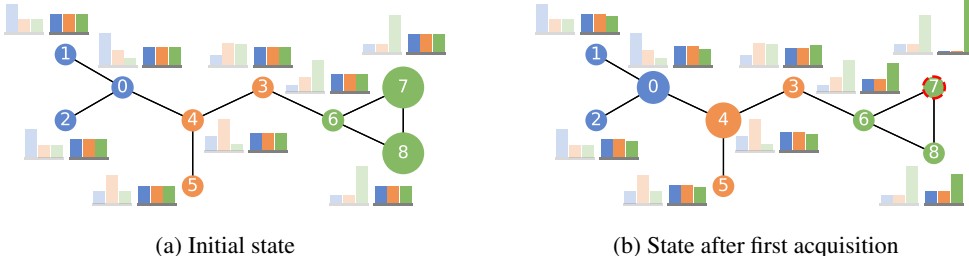

(a) Initial state           (b) State after first acquisition

Figure 12: Example of CSBM with 3 classes and 9 nodes. **Left**, we can see the initial state without any labeled node. On the **right**, we can see the state after we acquired node 7.

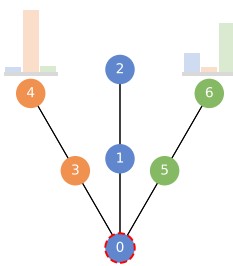

Figure 13: Aleatoric uncertainty is different for symmetric nodes when the affiliation matrix is not symmetric itself.

Figure 12b depicts the subsequent iteration following the acquisition of node 7. Due to the additional information introduced, node 8 exhibits reduced epistemic uncertainty. Consequently, the most promising nodes to consider next are nodes 0 and 4, as their total confidence remains relatively low, and they display higher aleatoric confidence than their neighbors. This is because both nodes connect to two nodes from the same class, bolstering their confidence levels.

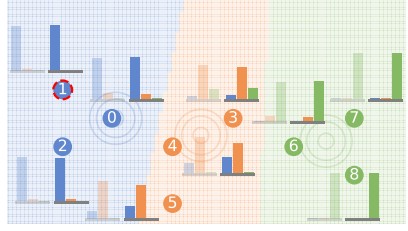 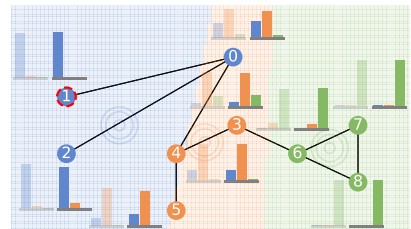

(a) Decision boundary given only the features       (b) Decision boundary given the features **and** structure

Figure 14: Example of CSBM with one labeled node (1). The shaded area represents the decision of the classifier. **Left**, only using the feature information. **Right** feature and structural information.

Figure 14a presents predictions made solely on feature information, with shaded regions and concentric circles demonstrating decision boundaries and class feature distributions, respectively. It is evident that incorporating feature information significantly increases confidence. Then, in Figure 14b, we adjust the position of node 0 away from the blue center and introduce the structural information. The shaded regions in this figure represent the prediction for node 0 and a noticeable shift to the right compared to Figure 14a can be observed. This shift demonstrates the impact of structural information on the classifier's decision boundary.

Finally, Figure 13 underscores the significance of modeling non-existing edges in uncertainty estimation. Despite their symmetric neighborhoods, the aleatoric uncertainty (without feature information) differs for nodes 4 and 6 as in the underlying CSBM model the green class is less likely to not associate with the blue class. This reveals that sensible uncertainty estimators need to also utilize the *absence of edges*.

## H  APPROXIMATING GROUND-TRUTH UNCERTAINTY ON REAL-WORLD DATA

**Approximating the Left-Hand Side (LHS)** We introduce a straightforward approach to instantiate the uncertainty disentanglement framework of § 5 by estimating both total and aleatoric confidence from GNNs that ought to fit the true underlying data generating distribution which is unknown for real-world datasets. Specifically, we train a SGC classifier $f_\theta$ on all labeled instances and treat its predictive distribution as approximation to $\text{conf}^{\text{total}}$:

$$\text{conf}^{\text{total}}(i, c) \approx f_\theta(\boldsymbol{A}, \boldsymbol{X})_{i,c} \tag{22}$$

Estimating the aleatoric confidence in a similar fashion requires access to the labels of unobserved instances $\mathbf{y}_{\mathcal{U}}$ which are not available in practice. Instead, we rely on $f_\theta$ to output pseudo-labels as an approximation $\hat{\mathbf{y}}_i \approx \text{argmax}_c f_\theta^*(\boldsymbol{A}, \boldsymbol{X})_{i,c}$. We then can model $\text{conf}^{\text{alea}}(i, c) \approx \mathbb{P}[\mathbf{y}_i = c \mid \boldsymbol{A}, \boldsymbol{X}, \mathbf{y}_{\mathcal{O}}, \hat{\mathbf{y}}_{\mathcal{U}-i}]$. Specifically, we can train a classifier $f_{\hat{\theta}}$ to output aleatoric confidences on both the true labels $\mathbf{y}_{\mathcal{O}}$ and pseudo-labels $\hat{\mathbf{y}}_{\mathcal{U}}$.

Note that for each node $i$, the set of unobserved pseudo-labels supplied to $f_{\hat{\theta}}$ differs as it must exclude node $i$ itself. This implies training $\mathcal{O}(n)$ models to approximate aleatoric uncertainty, which is intractable in many practical scenarios. Instead, we train one model $f_{\hat{\theta}}$ on all pseudo-labels at once, assuming the effect of removing a singular instance to be negligible. In practice, we observed there to be no substantial improvement in instead training one distinct classifier to approximate the aleatoric uncertainty of each node individually.

$$\text{conf}^{\text{alea}}(i, c) \approx f_{\hat{\theta}}(\boldsymbol{A}, \boldsymbol{X} \mid \mathbf{y}_{\mathcal{U}} = \hat{\mathbf{y}}_{\mathcal{U}})_{i,c} \tag{23}$$

Epistemic uncertainty is computed according to Def. 4. Again, since for an unlabeled node $i$ the label $\mathbf{y}_i$ is not known, we rely on the pseudo-labels to decide for which class to query epstemic uncertainty in US. That is, in each iteration we acquire node $v = \text{argmax}_i \text{conf}^{\text{epi}}(i, \hat{\mathbf{y}}_i)$.

**Approximating the Right-Hand Side (RHS)** An alternative approach is to not approximate epistemic uncertainty via total and aleatoric confidences but directly estimate the right-hand side of Thm. 1. To that end, the denominator can handily be approximated using the predictive distribution of a SGC $f_\theta$, similar to the aforementioned approximation of the left-hand side. The numerator, again, requires access to unavailable ground-truth labels $\mathbf{y}_{\mathcal{U}}$. Incorporation pseudo-labels in a similar fashion, we can estimate the numerator for each node $i$ and class $c$ separately. In contrast to approximating aleatoric uncertainty using pseudo-labels, where we have to fix all unobserved node labels at a fixed value at the same time, estimation of the right-hand side requires only one label at a time. Hence, we average over all pseudo-labels instead of picking the argmax.

$$\text{u}^{\text{epi}}(i) \approx \mathbb{E}_{c \sim f_\theta(\boldsymbol{A}, \boldsymbol{X})} \left[ \frac{\prod_{j \in \mathcal{U}-i} f_{\hat{\theta}}(\boldsymbol{A}, \boldsymbol{X} \mid \mathbf{y}_i = c)_{j, \hat{\mathbf{y}}_j}}{\prod_{j \in \mathcal{U}-i} f_\theta(\boldsymbol{A}, \boldsymbol{X})_{j, \tilde{\mathbf{y}}_j}} \right] \tag{24}$$

Here, the numerator is evaluated at the hard pseudo-labels $\hat{\mathbf{y}}_j = \text{argmax}_k f_{\hat{\theta}}(\boldsymbol{A}, \boldsymbol{X} \mid \mathbf{y}_i = c)_{j,k}$ predicted by the classifier $f_{\hat{\theta}}$ that is trained for each node $i$ and potential class $c$. The denominator probabilities are evaluated at the hard labels $\tilde{\mathbf{y}}_j = \text{argmax}_k f_\theta(\boldsymbol{A}, \boldsymbol{X})_{j,k}$ predicted by the classifier $f_\theta$. Note that similar to GEEM (Regol et al., 2020), approximating the RHS implies fitting $O(n * c)$ classifiers in each iteration at the benefit of making fewer approximative assumptions compared to modelling the LHS.

This approach bears some resemble to the GEEM strategy, as it averages over an estimate of the benefit each acquisition would provide. Its alignment with our theoretical results may explain its

success among the various strategies this work studies. Nonetheless, the proposed approximation to the right-had side of Thm. 1 differs from GEEM: (i) We estimate the probability of the classifier making correct predictions $\prod_{j\in\mathcal{U}-i} f_{\hat{\theta}}(\boldsymbol{A}, \boldsymbol{X} \mid \mathbf{y}_i = c)_{j,\hat{\mathbf{y}}_j}$ while GEEM estimates the risk $\sum_{j\in\mathcal{U}-i,k} f_{\hat{\theta}}(\boldsymbol{A}, \boldsymbol{X} \mid \mathbf{y}_i = c)_{j,k}$. (ii) Our framework weights the benefit against not acquiring a label $\prod_{j\in\mathcal{U}-i} f_{\theta}(\boldsymbol{A}, \boldsymbol{X})_{j,\tilde{\mathbf{y}}_j}$. Nonetheless, the similarity between both approaches is also reflected in their similar performance in Figs. 4 and 15a to 15d.

**Backbone Architecture and Training.** Similarly to GEEM, we employ an SGC model as the backbone classifier for our proposed framework. Effectively, SGC is a logistic regression model fit to diffused node features $\boldsymbol{X}$. We use the SAGA solver which is efficient for larger datasets to approximate the aleatoric uncertainty of the RHS as it uses pseudo-labels of all nodes, and we rely on liblinear in all other cases. Furthermore, we account for class imbalances and use a regularization weight of $\lambda = 1.0$. To mimic the GNNs used by other baselines, we diffuse the node features $\boldsymbol{X}$ for 2 iterations. While acquisition requires training and evaluating auxiliary models $f_{\hat{\theta}}$ on pseudo-labels $\hat{\mathbf{y}}$ or $\tilde{\mathbf{y}}$ for both approximation frameworks, we only train the underlying classifier $f_{\theta}$ that we report numbers on using ground-truth labels iteratively revealed by the oracle.

We also point out reasons for the proposed method to not be at its full efficacy yet due to the various assumptions and approximations we make. Specifically, different sources of error in estimating epistemic uncertainty can stem from (i) The classifiers $f_{\theta}$ and $f_{\hat{\theta}}$ not faithfully modelling the true generative process, as described in § 5, which results in suboptimal performance. (ii) The pseudo-labels $\hat{\mathbf{y}}$ not matching the true labels $\mathbf{y}$, which in turn leads to errors in approximating aleatoric confidence and querying epistemic confidence at the correct label. (iii) The classifiers being poorly calibrated, a tendency exhibited by some GNN architectures (Hsu et al., 2022). In fact, we observed similar experiments using a GCN instead of an SGC as a backbone to be unsuccessful. While both paradigms of approximating either the left-hand-side or right-hand-side of Thm. 1 aim to estimate the same quantity, they rely on different assumptions and approximations. Therefore, we expect them to behave differently in practice but both be indicative of the potential benefits of properly disentangling uncertainty.

Figs. 15a to 15d showcase the practical applicability of this framework. Similar to Fig. 4, we compare both approximations of the LHS and RHS to the best-performing US and non-US AL strategies. Approximating the LHS outperforms other strategies (except GEEM) on CoraML, while showing worse-than-random performance on co-purchase datasets. At the same time, the approximation of the RHS performs well on all datasets, also outperforming GEEM on PubMed.

From a theoretical perspective, both LHS and RHS approximations model the same quantity. If both were computed exactly, Thm. 1 guarantees that both strategies are equivalent. Nonetheless, they exhibit very different behaviors on some datasets when employing practical approximations: While on CiteSeer (Fig. 4) approximating the LHS performs superior, in all other cases the more costly approximation of the RHS shows higher efficacy. One potential reason for the weaker performance of approximating the LHS may be that aleatoric confidence is obtained from a classifier trained on many pseudo-labels. Especially when the accuracy of the backbone classifier is low, this may result in poor approximation qualities. In contrast, the LHS approximation only facilitates pseudo-labels when evaluating the gain of acquiring a certain label. This potentially allows for more flexibility, as even if the pseudo-label is incorrect, the assigned probability may still be similar enough to what the auxiliary classifier assigns to the unknown true label.

We want to stress that it is not the goal of our work to advertise this framework of uncertainty disentanglement as a novel state-of-the-art US strategy. Instead, we believe that the efficacy of applying our insights to a GNN that was neither designed for uncertainty estimation nor addresses any of the pitfalls related to modelling the data generating process mentioned in § 5 further verifies our theoretical claims and highlights their value in guiding the development of uncertainty estimation on graphs towards principled, highly effective AL. In particular, we believe this result to strongly motivate uncertainty estimation frameworks that either attempt to more faithfully approximate aleatoric and total uncertainty (RHS) or lifting the computational burden approximations of the LHS suffer from.

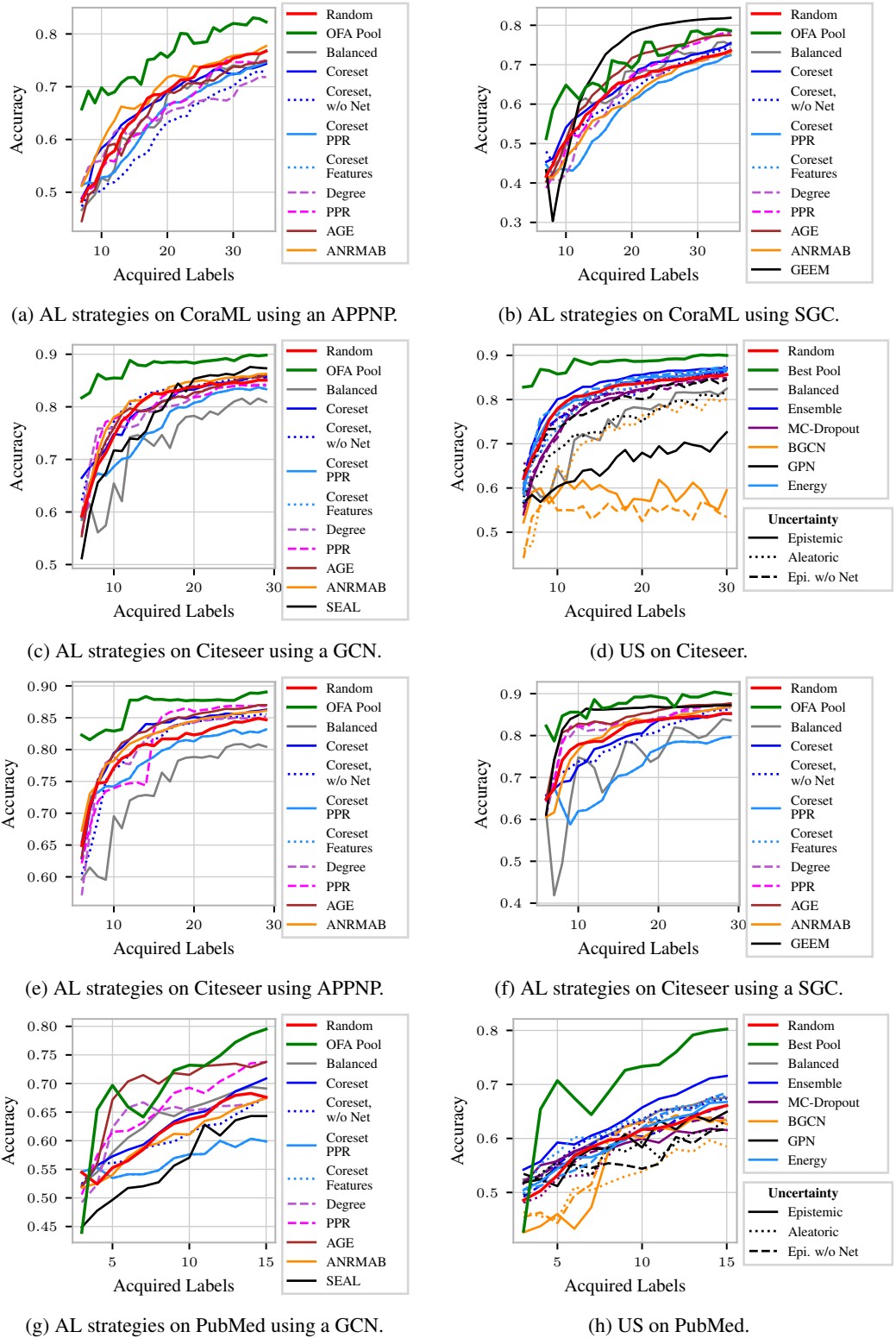

(a) AL strategies on CoraML using an APPNP.

(b) AL strategies on CoraML using SGC.

(c) AL strategies on Citeseer using a GCN.

(d) US on Citeseer.

(e) AL strategies on Citeseer using APPNP.

(f) AL strategies on Citeseer using a SGC.

(g) AL strategies on PubMed using a GCN.

(h) US on PubMed.

Figure 6: Accuracy curves of AL strategies, both non-uncertainty-based as well as US on different datasets for different models.

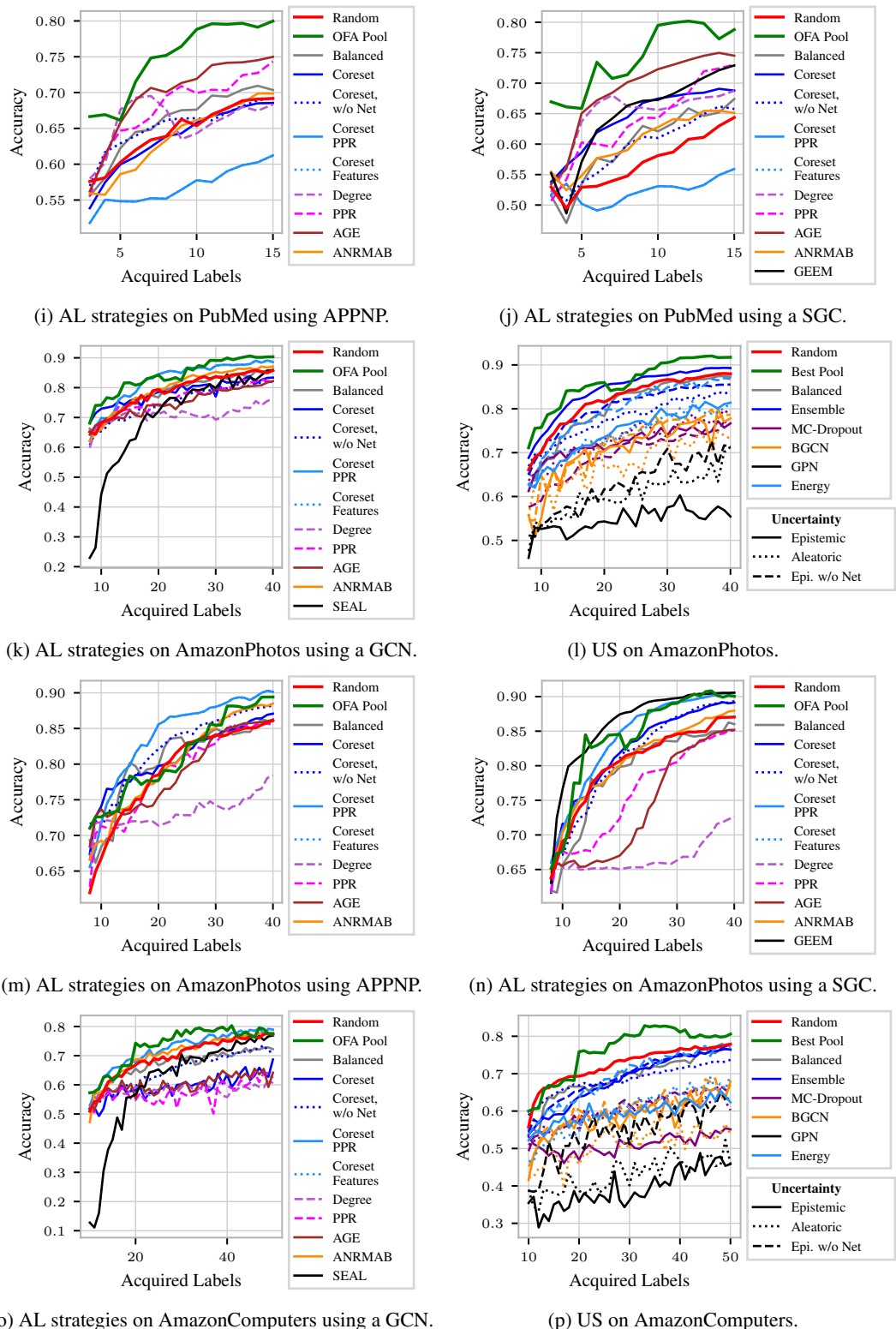

(i) AL strategies on PubMed using APPNP.

(j) AL strategies on PubMed using a SGC.

(k) AL strategies on AmazonPhotos using a GCN.

(l) US on AmazonPhotos.

(m) AL strategies on AmazonPhotos using APPNP.

(n) AL strategies on AmazonPhotos using a SGC.

(o) AL strategies on AmazonComputers using a GCN.

(p) US on AmazonComputers.

Figure 6: Accuracy curves of AL strategies, both non-uncertainty-based as well as US on different datasets for different models (cont.).

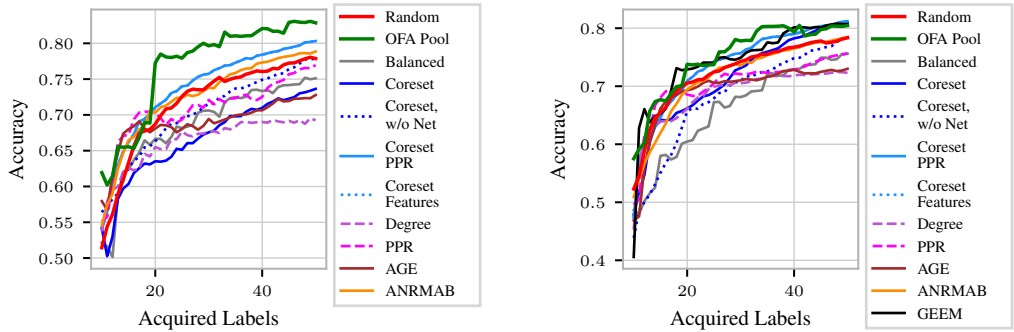

(q) AL strategies on AmazonComputers using APPNP.    (r) AL strategies on AmazonComputers using a SGC.

Figure 6: Accuracy curves of AL strategies, both non-uncertainty-based as well as US on different datasets for different models (cont.).

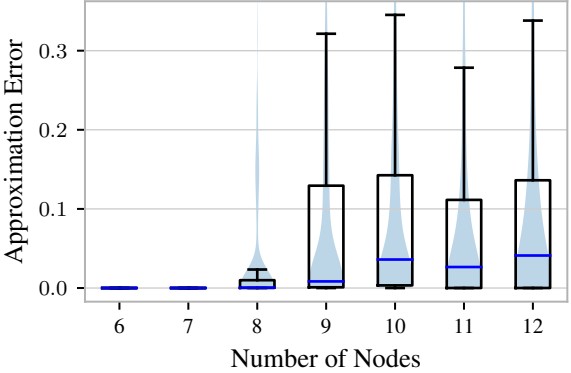

Figure 7: Absolute error distribution between approximate total confidence $q(\mathbf{y}_i)$ and true total confidence $p(\mathbf{y}_i \mid \mathbf{A}, \mathbf{X}, \mathbf{y}_{\mathcal{O}})$ for graphs of different sizes.

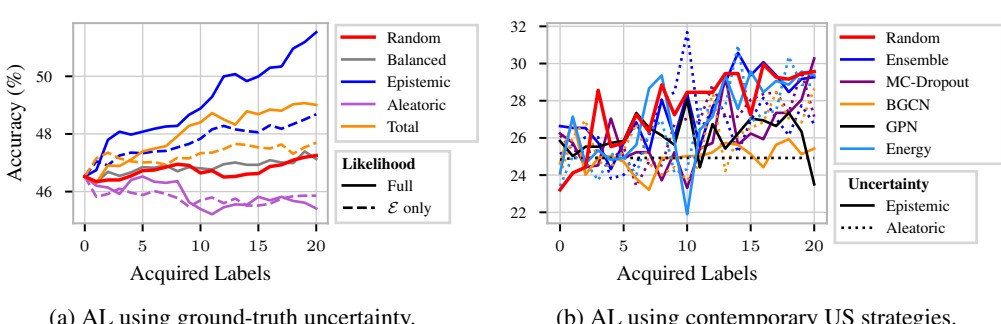

(a) AL using ground-truth uncertainty.    (b) AL using contemporary US strategies.

Figure 8: US on a CSBM with 1000 nodes and 4 classes. Ground-truth epistemic uncertainty significantly outperforms other estimators and random queries. Contemporary US can not outperform random sampling.

| Inputs | Model | OFA Pool* | Random | Balanced* | Coreset | AGE (A&X) | ANRMAB (A&X) | GEEM (A&X) | SEAL | Coreset PPR | PPR (A) | Degree | Coreset w/o Net | Coreset Inputs (X) | Epi./(Energy) (A&X) | Alea. (A&X) | Epi./(Energy) (X) | Alea. (X) |
|---|---|---|---|---|---|---|---|---|---|---|---|---|---|---|---|---|---|---|
| CoraML | GCN | 76.65±0.41 | 62.51±4.31 | 61.16±5.75 | 64.35±4.27 | 64.12±3.67 | 64.24±4.27 | n/a | 66.07±3.94 | 59.53±7.57 | 62.16±4.53 | 58.39±5.01 | 60.73±4.33 | 61.26±4.35 | 63.97±6.29 | 61.30±5.63 | 65.65±4.28 | 64.33±4.17 |
|  | APPNP | 76.10±0.22 | 67.72±5.70 | 65.39±5.11 | 67.32±4.20 | 66.12±3.36 | 69.49±5.52 | n/a | n/a | 71.04±6.28 | 65.52±3.48 | 64.57±4.79 | 62.01±5.26 | 64.49±4.38 | 64.92±7.02 | 67.68±5.09 | 69.59±4.28 | 66.69±5.03 |
|  | Ensemble | 77.98†±0.13 | 63.89±6.15 | 60.32±5.40 | 60.55±5.52 | 64.80±4.33 | 65.10±7.29 | n/a | n/a | 62.65±7.08 | 65.47±4.26 | 62.41±3.04 | 60.52±7.37 | 65.07±5.11 | 63.47±6.20 | 64.03±6.68 | 64.80±5.14 | 65.82±3.01 |
|  | MC-Dropout | 76.89±0.26 | 64.94±4.93 | 64.32±4.74 | 64.37±4.58 | 64.44±4.37 | 64.06±4.94 | n/a | n/a | 62.92±6.38 | 63.91±5.68 | 61.65±3.90 | 59.58±4.58 | 64.35±4.29 | 59.17±5.57 | 63.69±4.39 | 61.82±5.19 | 63.87±4.87 |
|  | BGCN | 58.42±1.81 | 45.76±3.26 | 45.98±3.90 | 49.37±3.47 | 51.25±4.48 | 47.23±3.04 | n/a | n/a | 39.43±3.87 | 54.64±2.54 | 48.68±3.68 | 43.65±3.73 | 44.85±3.15 | 44.45±2.28 | 46.61±3.07 | 42.74±3.57 | 48.11±3.16 |
|  | GPN | 62.66±1.02 | 56.50±4.73 | 56.69±2.88 | n/a | n/a | n/a | 71.39†±3.37 | n/a | 58.04±4.56 | 59.88±3.05 | 58.74±3.33 | n/a | 54.02±3.83 | 54.75±3.21 | 57.16±5.25 | 55.89±4.29 | 57.21±5.12 |
|  | SGC | 70.48±0.00 | 63.85±6.05 | 65.04±4.27 | 65.23±5.14 | 67.56±3.20 | 61.14±5.50 | n/a | n/a | 60.24±7.98 | 65.05±2.85 | 61.60±4.34 | 62.94±5.41 | 59.18±5.42 | 67.51±3.64 | 65.66±5.70 | 65.05±4.71 | 67.13±4.11 |
| Citeseer | GCN | 87.89±0.18 | 80.34±3.20 | 73.69±6.57 | 80.36±2.81 | 80.14±3.00 | 81.59±3.11 | n/a | 79.19±4.86 | 81.51±2.97 | 79.93±2.16 | 79.72±2.47 | 81.41±2.28 | 76.87±5.65 | 81.59±3.36 | 80.15±5.10 | 82.26±2.82 | 76.79±4.07 |
|  | APPNP | 86.81±0.12 | 80.77±5.34 | 74.43±8.03 | 82.62±2.53 | 83.00±2.59 | 82.31±3.10 | n/a | n/a | 80.70±4.49 | 81.37±4.85 | 81.36±3.32 | 80.89±5.26 | 79.20±7.16 | 80.37±6.14 | 77.77±8.04 | 81.94±4.95 | 79.15±4.96 |
|  | Ensemble | 88.24†±0.06 | 81.14±3.99 | 73.55±8.21 | 77.61±4.28 | 82.37±2.44 | 81.43±3.62 | n/a | n/a | 81.59±3.71 | 81.42±2.61 | 80.31±2.45 | 80.18±4.60 | 79.13±4.71 | 82.94±2.72 | 80.26±6.82 | 81.06±4.56 | 77.76±5.03 |
|  | MC-Dropout | 87.81±0.22 | 81.90±3.19 | 74.75±6.67 | 78.97±3.47 | 79.86±3.31 | 80.46±2.99 | n/a | n/a | 81.09±3.38 | 79.86±3.58 | 80.82±2.85 | 79.40±4.15 | 78.62±4.98 | 78.86±5.47 | 80.97±5.28 | 78.85±4.73 | 76.02±4.46 |
|  | BGCN | 73.00±1.75 | 70.17±3.64 | 61.72±5.20 | 73.86±2.14 | 76.30±1.42 | 70.70±3.28 | n/a | n/a | 69.32±3.41 | 73.77±2.73 | 75.55±1.47 | 68.59±3.91 | 67.51±3.64 | 58.68±7.96 | 71.23±3.80 | 54.85±6.96 | 70.33±4.14 |
|  | GPN | 80.85±0.69 | 77.07±5.06 | 68.18±7.28 | 79.29±4.21 | n/a | n/a | n/a | n/a | 72.83±6.88 | 79.52±2.53 | 80.65±1.69 | n/a | 71.33±7.55 | 65.31±10.21 | 74.28±7.63 | 78.73±5.24 | 77.28±4.85 |
|  | SGC | 87.63±0.00 | 81.04±5.06 | 73.70±6.74 | 79.38±4.44 | 84.21±1.95 | 81.03±3.28 | 85.25†±1.80 | n/a | 81.59±4.15 | 83.25±2.74 | 82.60±1.49 | 79.21±7.73 | 72.04±5.66 | 75.13±9.12 | 78.85±6.36 | 72.94±8.35 | 82.90±1.77 |
| Pubmed | GCN | 70.34±0.74 | 61.56±4.52 | 63.58±4.17 | 62.61±5.20 | 69.48±4.12 | 60.31±6.68 | n/a | 58.62±5.27 | 61.71±7.24 | 66.07±4.01 | 63.84±5.40 | 65.33±6.04 | 56.71±6.44 | 59.64±6.02 | 61.85±5.43 | 59.66±5.23 | 60.34±5.98 |
|  | APPNP | 75.09†±0.24 | 64.61±6.05 | 66.25±6.26 | 63.88±6.47 | 70.18±4.50 | 63.83±5.85 | n/a | n/a | 64.21±6.62 | 68.24±4.44 | 66.09±4.25 | 65.51±4.18 | 56.87±8.10 | 63.09±7.08 | 62.37±5.82 | 62.23±7.74 | 63.95±6.34 |
|  | Ensemble | 71.05±0.31 | 59.25±6.36 | 61.30±5.75 | 64.25±6.60 | 68.26±4.28 | 60.40±6.98 | n/a | n/a | 61.89±5.92 | 63.48±7.18 | 62.78±7.21 | 59.19±6.73 | 56.36±5.41 | 63.70±6.31 | 61.37±4.97 | 59.71±8.14 | 61.15±4.90 |
|  | MC-Dropout | 70.55±0.71 | 58.30±6.21 | 62.32±4.87 | 62.97±6.29 | 65.24±6.30 | 60.50±4.68 | n/a | n/a | 61.43±5.41 | 65.72±3.92 | 65.29±5.23 | 57.71±5.70 | 56.01±6.09 | 58.67±7.52 | 59.07±7.03 | 59.23±6.99 | 62.22±4.68 |
|  | BGCN | 65.82±1.64 | 53.59±5.46 | 54.99±4.78 | 59.29±4.21 | 56.93±3.14 | 52.68±2.92 | n/a | n/a | 53.40±4.46 | 59.12±2.49 | 55.93±5.47 | 53.63±4.50 | 51.40±4.15 | 55.19±3.83 | 52.81±4.77 | 57.09±2.93 | 54.62±5.29 |
|  | GPN | 73.35±1.20 | 59.59±6.46 | 61.37±6.34 | n/a | n/a | n/a | 64.82†±3.68 | n/a | 62.08±6.79 | 64.64±3.83 | 58.13±4.94 | n/a | 54.34±7.91 | 58.82±6.90 | 57.24±8.31 | 56.25±6.23 | 59.37±6.96 |
|  | SGC | 74.29±0.00 | 56.79±6.28 | 59.90±4.65 | 64.48±5.35 | 69.20±4.83 | 60.49±6.30 | n/a | 71.16±3.02 | 62.15±5.47 | 63.88±4.65 | 65.14±4.69 | 59.57±4.45 | 52.25±6.89 | 62.04±5.24 | 61.55±6.65 | 61.04±5.84 | 60.74±6.28 |
| AmazonPhotos | GCN | 84.37±0.41 | 79.06±3.86 | 78.72±3.59 | 78.58±2.44 | 75.17±5.50 | 79.97±4.20 | n/a | 75.40±4.41 | 70.20±8.55 | 76.30±3.13 | 71.00±5.92 | 82.31†±3.29 | 82.71†±3.03 | 79.72±6.19 | 74.66±7.12 | 79.96±4.84 | 79.63±4.59 |
|  | APPNP | 81.30±0.14 | 79.29±6.17 | 80.65±4.22 | 81.04±2.86 | 79.02±5.26 | 80.35±4.96 | n/a | n/a | 76.37±7.76 | 79.03±3.83 | 73.25±8.33 | 76.95±5.38 | 84.04±4.15 | 84.46±3.69 | 77.45±6.13 | 80.48±4.97 | 77.69±6.83 |
|  | Ensemble | 86.37±0.14 | 82.23±2.91 | 80.50±3.96 | 80.44±4.48 | 77.45±3.41 | 82.77±4.62 | n/a | n/a | 74.93±8.07 | 77.32±4.53 | 75.17±6.07 | 75.71±4.24 | 82.45±2.84 | 80.50±4.83 | 77.85±6.69 | 81.25±4.27 | 78.80±4.17 |
|  | MC-Dropout | 84.34±0.50 | 80.32±3.75 | 76.39±3.68 | 76.63±4.47 | 74.75±3.45 | 80.21±3.45 | n/a | n/a | 75.32±6.70 | 74.33±3.46 | 69.03±4.60 | 69.78±4.24 | 73.39±4.16 | 72.42±9.05 | 73.16±6.19 | 69.68±8.18 | 69.19±6.80 |
|  | BGCN | 75.24±1.35 | 71.22±3.86 | 70.19±4.50 | 67.15±4.70 | 65.69±5.06 | 70.69±3.38 | n/a | n/a | 59.34±7.49 | 64.51±4.24 | 61.23±5.84 | 70.83±3.36 | 73.39±4.16 | 70.83±3.36 | 67.83±5.83 | 72.21±2.84 | 69.19±6.80 |
|  | GPN | 56.85±2.12 | 62.80±4.22 | 60.69±3.49 | n/a | n/a | n/a | 86.43†±4.13 | n/a | 55.59±4.09 | 62.17±4.08 | 56.77±4.94 | n/a | 65.07†±2.74 | 54.78±3.57 | 60.53±4.17 | 62.90±2.61 | 62.41±2.75 |
|  | SGC | 84.46±0.00 | 80.52±4.89 | 79.24±5.79 | 82.32±2.58 | 74.01±5.96 | 80.92±4.04 | n/a | n/a | 66.94±7.86 | 66.16±4.06 | 66.63±8.12 | 81.47±5.46 | 84.24†±4.37 | 84.01±4.72 | 71.43±8.37 | 80.75±4.23 | 76.38±7.40 |
| AmazonComputers | GCN | 73.68±0.55 | 69.80±3.43 | 67.21±3.20 | 59.36±4.83 | 60.07±6.85 | 70.70±2.66 | n/a | 61.51±4.11 | 61.22±5.54 | 57.45±4.48 | 58.14±5.29 | 65.13±4.31 | 72.34±2.75 | 59.62±7.26 | 60.17±6.98 | 69.34±4.75 | 69.87±3.77 |
|  | APPNP | 76.92†±0.09 | 71.69±3.34 | 68.75±7.30 | 66.62±3.55 | 68.89±3.32 | 72.69±3.27 | n/a | n/a | 65.75±6.76 | 70.91±3.52 | 66.42±4.73 | 70.22±4.44 | 73.83±3.56 | 62.03±8.44 | 62.26±10.26 | 68.79±5.46 | 71.72±3.79 |
|  | Ensemble | 76.33±0.17 | 72.56±3.06 | 69.16±4.30 | 64.19±4.49 | 60.69±5.43 | 72.73±4.16 | n/a | n/a | 64.13±6.76 | 60.59±7.02 | 63.16±5.33 | 67.20±6.40 | 75.39±2.68 | 68.38±5.04 | 68.47±7.74 | 69.49±8.35 | 73.67±3.30 |
|  | MC-Dropout | 73.79±0.64 | 68.06±4.65 | 67.17±3.04 | 56.58±4.30 | 57.04±4.61 | 69.01±2.92 | n/a | n/a | 62.39±6.35 | 55.88±4.81 | 56.72±6.12 | 61.86±3.16 | 71.05±2.88 | 51.02±9.05 | 59.31±7.96 | 60.71±7.40 | 70.66±3.97 |
|  | BGCN | 65.38±2.01 | 60.52±3.72 | 60.72±2.86 | 45.65±3.02 | 46.72±3.74 | 60.32±4.31 | n/a | n/a | 39.86±8.46 | 43.61±4.17 | 45.52±2.93 | 58.85±3.41 | 60.79±3.89 | 58.64±5.96 | 51.11±6.23 | 60.19±3.82 | 59.20±3.65 |
|  | GPN | 53.90±2.37 | 52.26±5.90 | 49.65±7.06 | n/a | n/a | n/a | n/a | n/a | 34.71±4.90 | 49.30±5.00 | 45.21±5.32 | n/a | 56.32†±3.70 | 39.21±3.92 | 42.95±4.53 | 54.83±4.84 | 53.71±3.91 |
|  | SGC | 75.15±0.00 | 72.39±2.74 | 66.18±6.34 | 71.53±3.86 | 69.31±4.37 | 71.62±3.79 | 74.49†±3.43 | n/a | 58.52±0.44 | 70.35±3.84 | 68.70±3.24 | 68.14±5.46 | 73.91±3.30 | 61.02±5.74 | 59.34±10.50 | 66.53±5.58 | 69.24±5.52 |

Table 4: Average AUC (↑) for different acquisition strategies on different models and datasets and the corresponding standard deviation over 5 different dataset splits and 5 model initializations each. We mark the best strategy per model in bold and underline the second best. For each dataset, we additionally mark the overall best model and strategy with the † symbol.

Table 5: Average final classification accuracy (↑) for different acquisition strategies on different models and datasets and the corresponding standard deviation over 5 different dataset splits and 5 model initializations each. We mark the best strategy per model in bold and underline the second best. For each dataset, we additionally mark the overall best model and strategy with the † symbol.

| | | Baselines | | | | Non-Uncertainty A & X | | | | Non-Uncertainty A | | | Non-Uncertainty X | | Uncertainty A & X | | Uncertainty X | |
|---|---|---|---|---|---|---|---|---|---|---|---|---|---|---|---|---|---|---|
| Dataset | Model | OFA Pool* | Random | Balanced* | Coreset | AGE | ANRMAB | GEEM | SEAL | Coreset PPR | PPR | Degree | Coreset w/o Net | Coreset Inputs | Epi/(Energy) | Alea. | Epi/(Energy) | Alea. |
| CoraML | GCN | 81.32±0.00 | 72.80±3.74 | 71.29±6.64 | 72.95±4.17 | 72.54±3.61 | 74.64±3.46 | n/a | 73.08±6.13 | 69.33±6.04 | 71.10±4.26 | 67.31±5.93 | 72.36±4.27 | 71.21±4.18 | 71.34±5.39 | 70.69±7.56 | 75.59±2.82 | 74.08±3.79 |
| | APPNP | 82.27†±0.42 | 76.74±4.45 | 76.85±3.30 | 74.33±4.02 | 74.86±2.76 | 77.71±4.59 | n/a | n/a | 76.11±4.08 | 74.99±2.24 | 71.81±3.78 | 72.78±3.84 | 74.69±3.88 | 72.29±5.83 | 75.61±3.88 | 77.91±3.03 | 75.29±3.71 |
| | Ensemble | 82.14±0.37 | 74.25±4.81 | 72.38±4.57 | 68.32±3.58 | 73.82±2.05 | 75.40±4.56 | n/a | n/a | 72.76±4.57 | 72.96±2.64 | 69.36±3.30 | 69.77±5.69 | 74.12±3.77 | 74.98±3.58 | 75.51±3.48 | 74.63±3.64 | 75.62±2.29 |
| | MC-Dropout | 81.53±1.02 | 73.86±5.46 | 72.39±5.08 | 72.44±5.59 | 72.40±4.09 | 73.44±5.36 | n/a | n/a | 70.91±5.15 | 72.06±5.10 | 69.58±3.78 | 71.42±5.19 | 74.06±1.94 | 68.64±4.57 | 75.42±4.15 | 71.67±4.63 | 75.86±4.55 |
| | BGCN | 63.01±0.35 | 59.09±5.86 | 56.75±10.27 | 61.08±6.05 | 61.20±6.35 | 57.80±6.47 | n/a | n/a | 46.46±7.81 | 66.85±3.48 | 59.40±5.58 | 56.22±4.89 | 55.73±7.12 | 56.53±7.54 | 59.48±7.56 | 47.72±7.13 | 57.87±7.44 |
| | GPN | 68.29±5.01 | 64.24±5.57 | 62.60±5.90 | n/a | n/a | n/a | 81.89†±1.58 | n/a | 59.66±7.23 | 66.40±6.16 | 65.10±5.94 | n/a | 60.52±6.49 | 57.34±8.78 | 61.08±8.07 | 63.55±7.16 | 65.97±7.69 |
| | SGC | 78.57†±0.00 | 73.59±4.99 | 75.06±3.40 | 75.50±3.70 | 77.49±1.44 | 72.99±4.52 | n/a | n/a | 77.06±7.62 | 78.02±1.26 | 73.14±4.15 | 74.35±4.61 | 72.39±4.66 | 77.90±2.49 | 75.23±3.96 | 76.59±2.79 | 77.03±3.59 |
| Citeseer | GCN | 89.71±0.51 | 85.38±2.24 | 82.52±3.43 | 86.13±1.80 | 86.15±1.99 | 86.68±1.89 | n/a | 87.69†±2.00 | 86.03±1.72 | 84.31±1.58 | 84.31±1.59 | 85.51±1.90 | 84.01±1.56 | 86.74±2.45 | 86.39±2.89 | 87.21±1.98 | 87.39±1.95 |
| | APPNP | 89.12±0.23 | 84.82±3.78 | 81.69±5.31 | 85.96±2.64 | 87.09±1.55 | 86.09±2.08 | n/a | n/a | 85.31±3.32 | 86.99±1.48 | 85.33±1.68 | 85.78±2.98 | 82.97±5.03 | 86.55±3.81 | 84.49±5.34 | 85.83±3.38 | 87.27±1.90 |
| | Ensemble | 89.98±0.17 | 85.63±3.26 | 82.46±4.39 | 85.00±2.65 | 86.07±2.35 | 86.77±2.16 | n/a | n/a | 86.19±3.25 | 85.07±1.50 | 84.90±1.59 | 86.05±1.40 | 83.98±3.30 | 86.41±4.64 | 86.41±4.44 | 86.27±2.37 | 87.15±2.28 |
| | MC-Dropout | 89.55±0.56 | 86.64±1.90 | 83.00±4.37 | 86.52±1.88 | 85.50±1.83 | 86.07±2.77 | n/a | n/a | 86.01±2.24 | 84.56±1.95 | 85.23±1.13 | 85.12±2.47 | 84.74±3.07 | 85.63±2.50 | 87.59±2.09 | 84.84±3.57 | 88.06±2.00 |
| | BGCN | 77.72±5.98 | 80.96±4.76 | 67.20±9.09 | 80.45±3.60 | 79.59±4.58 | 79.37±4.66 | n/a | n/a | 75.14±6.50 | 80.44±4.52 | 81.05†±3.44 | 76.46±5.00 | 76.41±3.51 | 59.50±13.96 | 80.08±5.25 | 53.42±12.61 | 78.44±5.55 |
| | GPN | 82.97±3.37 | 82.57±8.73 | 75.72±7.46 | n/a | n/a | n/a | 87.33±1.95 | n/a | 77.06±7.62 | 83.35±2.24 | 82.46±2.34 | n/a | 79.14±4.86 | 72.61±7.77 | 81.87±5.27 | 84.58±3.86 | 81.60±3.65 |
| | SGC | 90.09†±0.00 | 85.45±3.72 | 82.86±3.35 | 85.26±2.90 | 87.84±1.10 | 87.09±1.86 | n/a | n/a | 87.17±2.14 | 87.49±1.05 | 86.73±1.73 | 86.37±2.99 | 80.23±4.86 | 75.63±10.15 | 86.19±3.54 | 80.16±10.35 | 87.78±1.98 |
| Pubmed | GCN | 79.51±1.60 | 67.60±5.10 | 69.12±4.02 | 70.87±5.94 | 73.79±3.92 | 67.44±8.55 | n/a | 65.01±7.30 | 66.73±5.80 | 73.72±3.72 | 67.57±6.48 | 67.66±5.04 | 59.87±7.29 | 66.75±8.22 | 67.84±6.27 | 68.46±7.13 | 65.62±7.53 |
| | APPNP | 79.97±0.51 | 69.18±5.57 | 70.36±4.76 | 68.54±7.37 | 74.98±4.48 | 69.86±5.37 | n/a | n/a | 69.19±5.81 | 74.33±3.52 | 68.33±4.29 | 69.24±5.44 | 61.23±9.30 | 70.86±5.61 | 68.16±6.96 | 68.09±6.57 | 68.93±6.57 |
| | Ensemble | 80.24†±0.56 | 66.10±6.96 | 67.54±5.33 | 70.89±6.16 | 72.65±3.98 | 68.74±6.24 | n/a | n/a | 68.24±5.75 | 69.44±6.61 | 66.11±6.92 | 65.03±5.24 | 60.25±6.79 | 71.57±6.75 | 67.35±5.07 | 66.10±6.61 | 68.07±5.96 |
| | MC-Dropout | 79.17±2.04 | 66.45±4.99 | 68.96±5.52 | 71.19±7.79 | 71.15±6.03 | 65.50±4.56 | n/a | n/a | 67.88±6.79 | 72.20±3.46 | 70.11±7.06 | 62.32±7.93 | 60.82±6.69 | 61.55±8.42 | 67.70±7.60 | 63.82±7.51 | 69.75±4.93 |
| | BGCN | 70.62±8.49 | 62.57±8.73 | 60.38±6.98 | 66.70±4.98 | 63.77±5.06 | 60.72±6.22 | n/a | n/a | 59.02±9.08 | 68.97±3.86 | 58.06±6.41 | 60.36±6.70 | 54.49±6.33 | 62.67±5.46 | 58.51±7.89 | 63.49±6.15 | 61.25±7.83 |
| | GPN | 77.61±4.23 | 63.75±8.43 | 66.06±8.02 | n/a | n/a | n/a | 72.91±5.03 | n/a | 66.96±7.94 | 67.79±5.93 | 61.50±6.74 | n/a | 54.32±9.25 | 64.94±7.58 | 63.31±8.19 | 61.74±6.70 | 65.38±9.24 |
| | SGC | 78.79±0.00 | 64.36±5.81 | 67.39±4.80 | 68.78±3.81 | 74.51±4.79 | 65.14±7.04 | n/a | n/a | 67.71±5.66 | 73.05±4.78 | 68.76±4.42 | 65.76±4.13 | 55.91±8.35 | 70.29±6.11 | 67.61±5.85 | 67.19±7.03 | 68.18±6.85 |
| AmazonPhotos | GCN | 90.34±1.30 | 85.76±3.16 | 85.76±2.53 | 83.37±2.57 | 82.20±2.98 | 87.07±3.40 | n/a | 85.92±2.50 | 75.73±8.56 | 82.28±3.15 | 76.27±5.99 | 81.76±3.61 | 88.58±2.45 | 81.39±7.57 | 81.12±5.41 | 86.82±4.51 | 84.73±4.33 |
| | APPNP | 89.41±0.38 | 86.14±4.30 | 86.01±2.46 | 87.07±1.60 | 86.24±2.13 | 88.49±2.49 | n/a | n/a | 81.93±7.54 | 85.52±2.33 | 78.44±6.60 | 88.24±1.60 | 90.14†±1.73 | 86.07±4.41 | 84.31±4.21 | 86.65±3.87 | 85.28±4.15 |
| | Ensemble | 91.74†±0.33 | 87.97±2.14 | 87.29±2.26 | 84.82±3.48 | 83.32±4.04 | 88.10±3.30 | n/a | n/a | 79.82±7.86 | 83.36±4.01 | 78.60±5.09 | 83.51±3.15 | 89.96±1.61 | 89.28±2.36 | 83.92±6.09 | 85.53±4.31 | 87.17±3.75 |
| | MC-Dropout | 90.71±1.31 | 86.04±3.33 | 84.13±4.42 | 83.37±3.20 | 80.08±7.52 | 87.53±3.21 | n/a | n/a | 79.07±6.75 | 80.45±9.57 | 72.42±6.13 | 82.11±6.62 | 88.85±6.26 | 76.69±8.01 | 78.69±5.88 | 76.44±4.43 | 87.09±2.69 |
| | BGCN | 82.67±1.22 | 79.22±7.13 | 79.07±7.55 | 69.90±9.01 | 73.08±9.53 | 82.32±4.67 | n/a | n/a | 59.80±14.70 | 69.34±10.50 | 62.35±12.45 | 76.64±6.02 | 72.46†±10.43 | 77.81±8.13 | 73.15±9.69 | 78.66±8.01 | 77.17±7.59 |
| | GPN | 64.54±7.29 | 64.59±8.87 | 66.42±9.09 | n/a | n/a | n/a | 90.57†±2.85 | n/a | 56.95±9.79 | 71.18±9.41 | 59.87±9.57 | n/a | n/a | 55.44±7.09 | 71.29±8.57 | 71.33±9.67 | 70.29±11.14 |
| | SGC | 90.03±0.00 | 87.04±2.72 | 86.01±4.21 | 89.13±1.79 | 85.20±2.79 | 87.95±2.43 | n/a | n/a | 72.18±8.94 | 85.08±2.70 | 72.66±6.71 | 89.34±2.13 | 90.52±1.51 | 90.22±2.48 | 78.75±7.06 | 85.71±3.10 | 84.25±5.51 |
| AmazonComputers | GCN | 77.46±3.68 | 77.25±3.74 | 72.22±5.26 | 68.69±7.28 | 64.17±9.53 | 77.93±2.95 | n/a | 76.92±2.00 | 67.04±6.67 | 62.32±9.62 | 60.49±8.10 | 71.51±4.11 | 78.94†±4.69 | 62.35±11.46 | 69.54±8.69 | 77.08±4.57 | 77.84±3.28 |
| | APPNP | 82.86†±0.25 | 77.87±2.37 | 75.12±4.60 | 73.64±4.12 | 72.78±4.66 | 78.86±3.23 | n/a | n/a | 70.73±6.69 | 76.96±3.02 | 69.24±5.44 | 77.85±2.75 | 80.31±2.32 | 67.25±9.27 | 71.19±10.07 | 78.44±4.37 | 79.87±3.42 |
| | Ensemble | 80.58±0.98 | 77.88±2.90 | 76.50±3.81 | 69.20±5.50 | 66.15±4.00 | 78.79±4.64 | n/a | n/a | 71.33±3.58 | 65.49±5.88 | 66.32±5.77 | 71.42±5.38 | 80.50±0.38 | 76.41±3.94 | 73.71±7.31 | 77.38±5.33 | 80.38±4.37 |
| | MC-Dropout | 78.70±3.03 | 75.86±3.93 | 74.58±2.70 | 63.65±11.06 | 62.67±8.51 | 75.07±3.73 | n/a | n/a | 69.69±6.56 | 59.78±11.42 | 58.54±7.70 | 72.24±4.90 | 78.17±3.71 | 55.12±11.72 | 60.21±11.38 | 65.25±10.33 | 77.00±3.59 |
| | BGCN | 69.00±12.27 | 63.23±9.50 | 67.44±7.69 | 39.84±15.30 | 45.25±13.93 | 65.87±9.88 | n/a | n/a | 35.80±15.73 | 45.34±12.66 | 44.23±12.92 | 63.64±7.52 | 62.32±13.64 | 67.87±10.76 | 53.71±13.66 | 67.06±8.51 | 64.32±11.97 |
| | GPN | 58.06±11.37 | 59.89±13.72 | 58.45±11.58 | n/a | n/a | n/a | 80.69±2.65 | n/a | 32.00±8.75 | 60.63±10.59 | 46.42±16.56 | n/a | 64.79±0.00 | 45.85±13.64 | 45.95±14.32 | 63.25±5.79 | 64.24±6.71 |
| | SGC | 80.46±0.00 | 78.35±2.44 | 75.59±3.07 | 80.98±2.86 | 73.01±4.68 | 78.49±2.34 | n/a | n/a | 64.92±5.75 | 75.69±3.56 | 72.31±3.54 | 78.24±3.16 | 81.19†±2.25 | 68.70±8.90 | 67.44±10.26 | 73.46±6.06 | 78.38±4.05 |

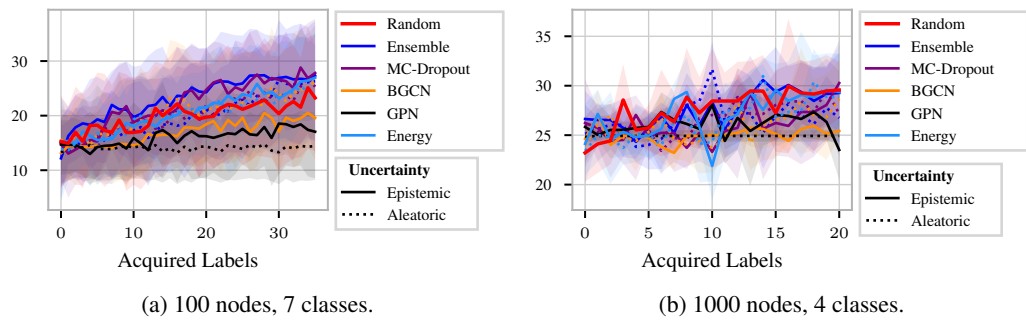

(a) 100 nodes, 7 classes.

(b) 1000 nodes, 4 classes.

Figure 9: Average performance of US on CSBMs with 100 and 1000 nodes respectively including standard deviations over 5 independent samples from the generative process.

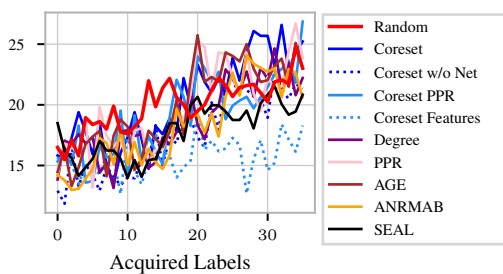

Figure 10: Performance of traditional AL strategies on a CSBM with 100 nodes and 7 classes.

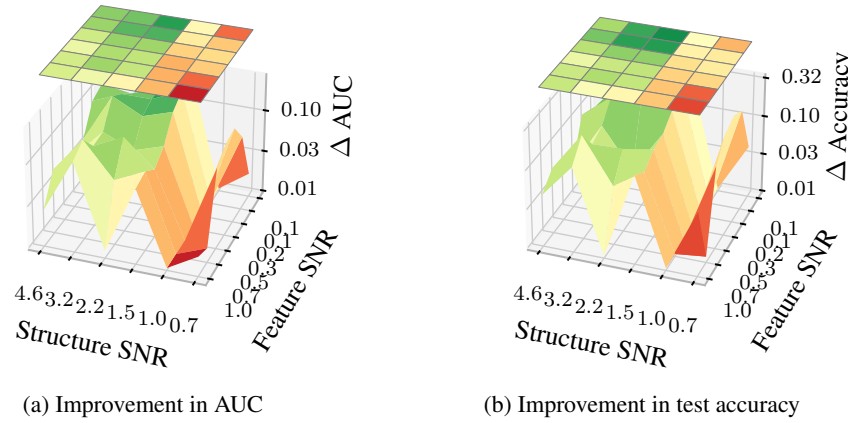

(a) Improvement in AUC

(b) Improvement in test accuracy

Figure 11: Evaluating the absolute improvement of Uncertainty Sampling using epistemic uncertainty over random acquisition for different structure and feature SNRs.

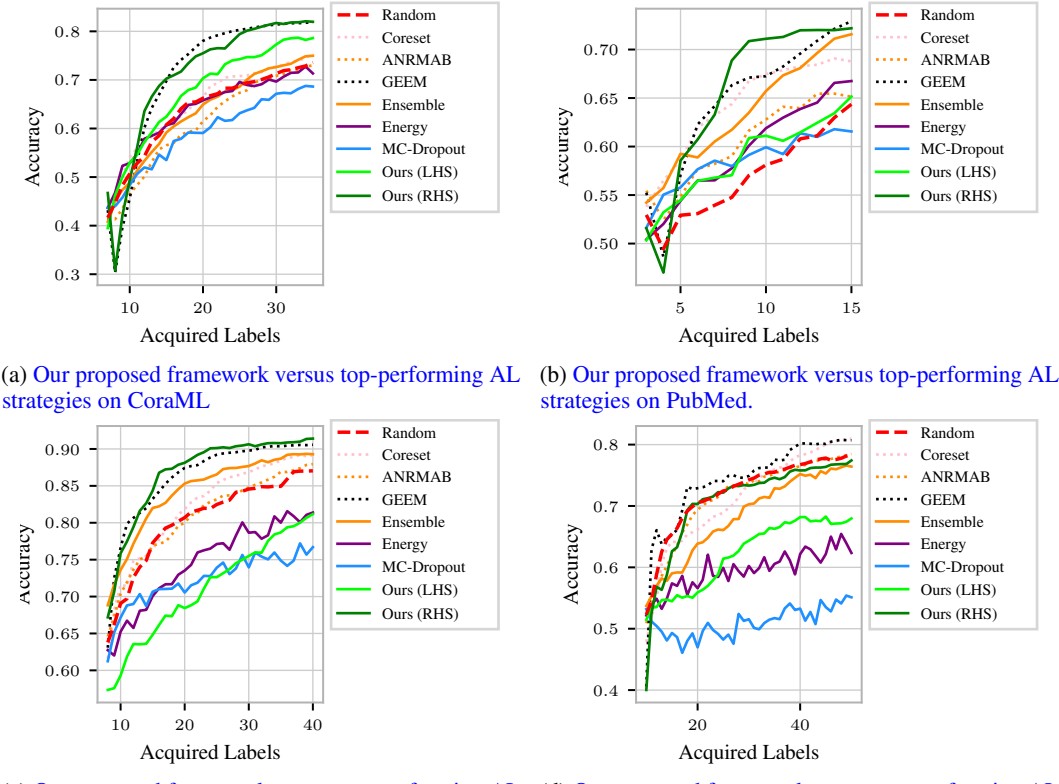

(a) Our proposed framework versus top-performing AL strategies on CoraML

(b) Our proposed framework versus top-performing AL strategies on PubMed.

(c) Our proposed framework versus top-performing AL strategies on Amazon Photos.

(d) Our proposed framework versus top-performing AL strategies on Amazon Computers.

Figure 15: Our proposed uncertainty disentanglement framework applied to a SGC classifier both estimating the left-hand side and right-hand side of Thm. 1.

