# OpenReview forum: "Uncertainty for Active Learning on Graphs"
_ICLR.cc/2024/Conference — Submitted to ICLR 2024_

### Official Review · Reviewer_FXsp · 2023-10-28

**Soundness:** 2 fair
**Presentation:** 3 good
**Contribution:** 1 poor
**Rating:** 3
**Confidence:** 3

**Summary:**

This paper presents a comprehensive study of applying Uncertainty Sampling (US) within the Active Learning (AL) framework for node classification within graphs. The authors provide a benchmark for AL and evaluate the performance of AL baselines using real word datasets. Additionally, they propose novel Bayesian uncertainty estimation methods based on the ground truth labels, and illustrate their effectiveness using synthetic CSBM dataset.

**Strengths:**

The paper offers a thorough evaluation of AL performance through a series of well-conducted experiments and a qualitative analysis.

**Weaknesses:**

The proposed ground truth uncertainty is not so useful and the uncertainty sampling US based on it is not practical. During prediction procedure, the ground truth label remains unknown and therefore it is inappropriate to define an uncertainty based on it.

US with knowledge of ground truth label would benefit from the information leakage and so the good performance in the CSBM dataset is not achievable in real-world datasets. For example, in traditional AL algorithms, it's difficult to select a node for query when the classifier gives the ground truth label of the node a low predictive probability, although querying such node would provide the classifier a lot information. Take, for instance, a scenario where  p(ground truth class| y_i ) = 0.1 and p(incorrect class| y_i ) = 0.9. Typically the prediction to incorrect class of y_i might be considered confident and AL algorithm will not choose y_i for querying, and such error will cause general AL algorithm not perform as good as random sampling. But for US based on ground truth uncertainty, the epistemic uncertainty is large and the node will be selected. Therefore, the good performance in the CSBM dataset is not practical.

**Questions:**

Please explain the practical application of the ground truth uncertainty.

---

> ### Author Response · Authors · 2023-11-16
>
> We want to thank the reviewer for their valuable time and feedback to review our manuscript.
>
> The reviewer is correct that the ground-truth uncertainties actually use the ground-truth labels and this technically implies data leakage. However, our argument goes for the existence of such optimal (oracle-like) uncertainty measures and not utilizing them in a practical context.
>
> Our main goal is to study the link between US and AL. Our theoretical result therefore implies that using the optimal epistemic uncertainty (which, again, can not be computed exactly in practice) is an optimal AL strategy. We first supply this with an empirical study on CSBMs under the idealized setting of full access to ground-truth uncertainties. This allows us to study the effects of disentangling uncertainty in an isolated fashion. We do not intend to propose ground-truth uncertainty as an AL strategy on CSBMs. We also want to point out that our theoretical analysis still fully translates to any dataset, albeit that ground-truth uncertainty, while also existing in this context, is not available and needs to be approximated instead. In our revised manuscript, we also accommodate learning the generative process (e.g. using a GNN) into our theory while retaining all optimality guarantees.
>
> We agree with the concern whether the strong performance in an ideal setting (CSBMs) does imply a practically usable AL strategy. Therefore, to further strengthen the claim that our findings translate to real data, we also added an experimental study on real-world graph datasets. Here, we used simple GNN-based approximations to ground-truth uncertainty (see Appendix H for details) and did not facilitate any ground-truth labels (i.e. data leakage) hence being a practical approach to AL. We find that disentangling uncertainty according to our theory already outperforms many SOTA methods and US methods on most datasets (see Figures 6 and 15). We believe that this strongly underlines that our findings also translate well to realistic settings and can inform US strategies towards aligning with AL.
>
> While the ground-truth uncertainty is defined with respect to information that is not available in practice (data generating process, labels of unobserved nodes), we empirically verify that trying to approximate them according to our theoretical framework gives a very strong AL strategy off-the-shelf. We believe that this shows that our theoretical analysis is highly relevant to the field of AL on graphs and can guide the development of novel US methods in a theoretically sound and well-motivated fashion.
>
> We again want to point out that regardless of outperforming other AL strategies on multiple datasets in a realistic setting without data leakage, the core contribution of our work is its theoretical analysis. Understanding if and how UQ relates to AL bridges an important aspect of uncertainty modeling neglected by previous advances in the field of UQ. The efficacy of an approximative application of our insights is just one instance of how our work can impact the development of UQ methods. For example, future work could aim at better modeling the unknown data-generating process, e.g. by considering non-edges. We also make a case for considering AL when evaluating the effectiveness of novel UQ strategies on graphs. Consequentially, we updated our manuscript to highlight the contribution and goal of our paper more distinctly (see Sections 1 and 6).
>
> ## Final Remarks
>
> We hope that the addition of a practical, novel method together with an experiment on real-world data without data leakage that outperforms SOTA US approaches convinces the reviewer that our analysis poses a significant and valuable, practical contribution and is worth sharing with the scientific community. If so, we kindly ask the reviewer to reconsider their score.

---

> > ### Author Response · Authors · 2023-11-21
> >
> > In light of the end of the author-reviewer discussion period on Wednesday, we would again like to kindly highlight our response and the manuscript's revised sections. We hope that we adequately address your concerns and are interested in your thoughts. If there are any outstanding or further questions we are delighted to discuss these.

---

> > > ### Comment · Reviewer_FXsp · 2023-11-21
> > >
> > > I appreciate your thoughtful response to my feedback. After careful consideration of your arguments and the points raised in your rebuttal, I maintain my score for your paper.

---

### Official Review · Reviewer_KZ6P · 2023-10-31

**Soundness:** 2 fair
**Presentation:** 2 fair
**Contribution:** 2 fair
**Rating:** 3
**Confidence:** 4

**Summary:**

The authors establish a benchmark for uncertainty sampling based active learning approaches for graph data. The paper also proposes a Bayesian uncertainty estimation to actively select the node. This estimation is based on the knowledge of data-generating process. The authors validate the effectiveness of their approach with both theoretical analysis and empirical experiments.

**Strengths:**

a. This paper studies the active learning problem with graph data from an interesting perspective--uncertainty sampling strategy and propose a new Bayesian uncertainty estimation.

b. The authors provide both theoretical analysis and empirical results to show the effectiveness of the proposed estimation.

**Weaknesses:**

a. Theoretical contributions in this paper appear to be somewhat limited. The proposed uncertainty estimation is based on the posterior probability given the ground-truth label of the unobserved nodes. However, the essence of active learning lies in addressing this problem without access to ground-truth information, which remains inadequately addressed.

b. The experimental results provided are restricted to synthetic data, and the method's reliance on knowledge of the true data generation process probabilities pose practical challenges. How to approximately compute the estimation remains unclear.

c. In the empirical evaluation, the compared baselines are only random queries and other uncertainty-based methods, the state-of-the art methods are missing, e.g. SEAL[1] and IGP[2].

d. The paper's presentation could be improved. For instance, when introducing concepts like aleatoric and epistemic uncertainty, the authors provide limited explanations and intuitions, potentially causing readers unfamiliar with these terms to struggle to follow the paper.

[1] Li Y, Yin J, Chen L. Seal: Semisupervised adversarial active learning on attributed graphs[J]. IEEE Transactions on Neural Networks and Learning Systems, 2020, 32(7): 3136-3147.
[2] Zhang W, Wang Y, You Z, et al. Information Gain Propagation: a new way to Graph Active Learning with Soft Labels[J]. arXiv preprint arXiv:2203.01093, 2022.

**Questions:**

a. How can the proposed uncertainty estimation be computed in practical scenarios where true data generation probabilities are unknown? Are there methods or approaches to approximate this estimation without relying on ground-truth knowledge?

b. What's the performance of non-US based active learning methods on CSBMs?

---

> ### Author Response · Authors · 2023-11-16
>
> We thank the reviewer for the time and the thorough review. We are delighted that the reviewer agrees with us, that we study the AL problem from an interesting perspective.
>
> ## Missing methods
>
> **Addressing weaknesses a and b:**
> We understand the concerns the reviewer has about the applicability of our results to a practical setting in which neither all labels nor the generative process is available. Our analysis, however, is purely theoretical: We show that ground-truth epistemic uncertainty is an optimal AL strategy. This result about the alignment of uncertainty estimation and AL also holds on real graphs, even though the proposed ground-truth uncertainties can not be computed exactly in practice. Our goal is to formally justify why we should expect US sampling to be an effective strategy in the first place. Furthermore, we can formally prove that disentanglement plays a crucial role in effective US, which again translates to real-world scenarios. Our work therefore provides a sound, theoretical basis on which novel US can be developed and applied to AL for interdependent data, which we believe to be a valuable contribution to the field.
>
> We want to point out that it is not the access to ground-truth data that is central to the effectiveness of epistemic US: Measures of total, aleatoric, and epistemic uncertainty are defined in terms of ground-truth information (see Defintions 1-3) and exhibit very different efficacy to AL.
>
> Nonetheless, we agree that our manuscript benefits from a more practical evaluation as well. To that end, we extended the theoretical framework to also allow a classifier to model the parameters of an underlying process in a learnable fashion, a framework that GNNs for example fit into. Based on this, we apply our theoretical insights to real-world data using GNNs as an approximative approach to our proposed, unknown ground-truth uncertainties. In this evaluation, we do not utilize any information about unavailable labels or the generative process. Nonetheless, this very off-the-shelf application of our findings elevates the uncertainty of a GNN classifier to outperform SOTA methods on multiple datasets. We hope to specifically also address weakness b with this additional empirical study, as we now propose an approximate estimator of epistemic uncertainty that aligns with our theory. We additionally discuss potential approximative errors of this simplistic approach in Appendix H.
>
> Overall, we also adapted the manuscript to address the overall intention of our analysis more clearly: Our goal is not the proposal of a novel AL strategy, but rather to bridge the gap between principled US and AL on graphs. We theoretically prove that disentangled US and AL align in general. We reveal pitfalls in UQ and how to address them with an evaluation in an idealized setting on a CSBM. Lastly, we confirm the applicability to real-world scenarios by evaluating a simple approximative realization of our insights on real graphs and observing strong performance.
>
> **Addressing Question a:** Estimating the true generative process of graph data is indeed a challenging topic. The aforementioned approach utilizes a simple GNN to address this problem and exhibits already reasonable performance. Nonetheless, we believe that our work can guide US toward the development of more faithful estimators: For example, on CSBMs, we observed non-edges to play a crucial role in successful US. Contemporary GNNs do not explicitly utilize such information which we conjecture to be one interesting direction for future research.

---

> > ### Author Response · Authors · 2023-11-16
> >
> > ## Other Baselines
> >
> > **Addressing Weakness c:**
> > We thank the reviewer for the interesting pointer to these two methods. SEAL is a very interesting approach and we have run experiments using SEAL and included them in all benchmarking Figures and Tables in our updated manuscript. As one can see, it provides merit on some graphs while not outperforming random acquisition on others. However, tuning the hyperparameters of SEAL might improve performance even further which we would include in the camera-ready version of the paper. These results strengthen our point further that non-UQ-based AL methods can outperform random queries while US falls short of this so far.
> >
> > The IGP approach deals with a very interesting setting. Assuming soft labeling is cheaper, they query the oracle with the predicted label and retrieve the answer as to whether or not the prediction was correct without acquiring its true label. We included a description of this setting in our revised related work section since we believe it is interesting to the readers. However, their setting is not directly applicable to an AL scenario with an oracle that returns the true label of an instance which is what most literature in AL on graphs discusses.
> >
> > While those two approaches do not fundamentally change any of our results in the benchmark or analysis, we think including them in our paper improves the manuscript, and thus we are thankful to the reviewer for pointing them out.
> >
> > ## Presentation:
> >
> > **Addressing Weakness d:** We allocated more explanation to the aleatoric and epistemic uncertainty notions to the background section 2, including an intuitive example to better help readers familiarize themselves with the concepts of the topic (see the updated version of our manuscript). If you have any concerns about the paper’s presentation, we are happy to address them as well.
> >
> > **Question b:**
> >
> > Our analysis on CSBMs is not purposed as a benchmark for AL strategies in practice. Instead, our goal is to empirically confirm the validity of our theoretical results and exemplify the importance of disentangling uncertainty. Nonetheless, we agree that benchmarking conventional AL strategies against the CSBM is an interesting additional experiment and added a corresponding plot in the Appendix (Figure 10).
> >
> > ## Final comments
> >
> > We again want to thank the reviewer for their time and valuable feedback. We hope that our revised manuscript can convince them that our analysis translates well to real-world AL on graphs and that while not proposing a novel AL strategy, our theoretical and empirical insights into the general problem including an extensive benchmark close a highly relevant gap in the literature and therefore are a valuable contribution to the field worth publishing. If so, we kindly ask the reviewer to reevaluate their score.

---

> > > ### Comment · Reviewer_KZ6P · 2023-11-21
> > >
> > > Thank you for your response. I'm interested in understanding more about the auxiliary classifiers used in the approximation method. Could you provide additional details regarding these auxiliary classifiers, such as their architecture or training process? Additionally, regarding the baseline methods, I'm curious whether they also utilize pseudo-labels from the auxiliary classifiers for training. Would the performance of these baselines benefit from using pseudo-labels? Furthermore, while I noticed that the results for Citeseer are included in the main content, I found more detailed results in Appendix H. However, I couldn't find sufficient text description accompanying the figures in Appendix H. Could you please provide more elaboration on what these figures represent and demonstrate?

---

> > > > ### Author Response · Authors · 2023-11-21
> > > >
> > > > We want to thank the reviewer for reviewing our updated work and the time allocated for that!
> > > >
> > > > ## Usage of Pseudo-Labels
> > > >
> > > > The framework we propose does not utilize auxiliary labels during training but only for acquiring new node labels. We report the accuracy of both our proposed framework and the top-performing baselines after re-training exclusively on node labels that the oracle provides and no-pseudo labels. We added clarification regarding this to Appendix H.
> > > >
> > > > **Baselines.** It is not straightforward how all other baseline strategies would make use of such pseudo-labels for the acquisition. However, we can argue that both Monte-Carlo (MC) approaches (Dropout, Ensemble, BGCN) and GEEM already make use of pseudo-labels to some extent when acquiring new nodes:
> > > >
> > > > - MC approaches to utilize the variance of predictions for the class $\hat{c}$, i.e. the pseudo-label of this class.
> > > > - GEEM explicitly uses pseudo-labels and auxiliary classifiers: The risk estimation they propose evaluates the probability that the auxiliary classifiers assign to the class $\hat{c}$ they predict for each node, i.e. the pseudo-label.
> > > >
> > > > **Our Framework.** We make use of pseudo-labels as follows:
> > > > - Approximating the RHS: We train one auxiliary classifier on all pseudo-labels to approximate aleatoric uncertainty.
> > > > - Approximating the LHS: We train multiple auxiliary classifiers to approximate the numerator of the LHS for each instance individually.
> > > >
> > > > In both frameworks (RHS and LHS), we need to also evaluate probabilities at the true labels of unobserved nodes, which are not available and thus also approximated to be the pseudo-labels of the classifier. After using pseudo-labels to select the node label to acquire and query the oracle, we retrain the classifier exclusively on real labels. Thus, the accuracy curves we report both for baselines and our framework only show models that are not trained on any pseudo label.
> > > >
> > > > ## Details on the Auxiliary Classifier
> > > >
> > > > As we are committed to providing good understanding for all readers, we are thankful for the reviewer to point out architectural aspects that are not clear. We added all aforementioned points as well as details about the backbone classifier and its training to Appendix H. We use an SGC mainly for two reasons:
> > > > - It is fast to train, allowing us to approximate the LHS which requires the training of many auxiliary classifiers.
> > > > - We empirically found it to give more stable (and potentially better calibrated) probabilities than a GCN. It is crucial for our proposed framework that the predicted probabilities represent the true belief of the classifier $f_\theta$ as faithfully as possible, as our acquisition function compares probability ratios.
> > > >
> > > > We added a thorough description of this point to Appendix H as well. Note that just using an SGC classifier alone does not consistently yield better US performance (see Tables 4 and 5), indicating that the performance increase should be attributed to our proposed framework.
> > > >
> > > > ## Discussion of Figure 15
> > > >
> > > > Lastly, as requested by the reviewer, we also provide an in-depth discussion about Figure 15, comparing our framework to other well-performing US and non-US methods as well as elaborating on the performance of both LHS and RHS approximations in relation to our theoretical findings and its practical implications.
> > > >
> > > > ## Final Remarks
> > > > We highlight additional changes to the manuscript in Appendix H in light blue. If any further requests or unclear points remain, we are happy to address them accordingly as well. If the reviewer is satisfied with our explanation and updated manuscript, we would be glad if they would consider raising their score.
> > > > We again want to thank the reviewer for taking the time to engage with us in this extended and fruitful discussion.

---

### Official Review · Reviewer_5NFL · 2023-10-31

**Soundness:** 2 fair
**Presentation:** 3 good
**Contribution:** 2 fair
**Rating:** 5
**Confidence:** 4

**Summary:**

This work is an empirical study of a typical Active Learning method, Uncertainty Sampling (US) for node classification on graphs. The authors present an extensive benchmark for US methods that goes beyond predictive uncertainty, revealing that, the US employing modern uncertainty estimators struggles to outperform random queries consistently. The authors establish ground-truth Bayesian uncertainty estimates for a Bayesian classifier based on the underlying graph generative process, providing formal evidence of the alignment between US and AL. When they apply their approach using a Clustered Stochastic Block Model (CSBM), they empirically confirm the effectiveness of US when uncertainty estimates are accurately disentangled into aleatoric and epistemic uncertainty while considering all available graph information.

**Strengths:**

- This work provides an empirical study for US with node classification on graphs, highlighting both its efficacy and potential limitations.

- An important finding is that the existing AL methods cannot outperform random sampling benchmarks.

**Weaknesses:**

- The study primarily concentrates on a specific graph type, the CSBM, which might not fully represent the characteristics of all real-world graphs.

- Novelty concern: undoubtedly, this work offers an extensive exploration of uncertainty-based Active Learning (AL) within the context of graphs. However, it does not introduce any novel methods for active learning in the graph domain.

**Questions:**

- In the experimental results, such as Figure 3, the curves depicting acquired labels versus accuracy exhibit significant fluctuations. Did the authors conduct repeated trials to mitigate these fluctuations in the model's performance?

- In the Introduction section, the authors dedicated an extensive portion of the text to explain uncertainty sampling. This level of detail might be excessive as uncertainty sampling is a straightforward concept. It would be more beneficial to present the essential formulations and allocate additional space to elaborate on active learning in graph-related tasks.

---

> ### Author Response · Authors · 2023-11-16
>
> We want to thank the reviewer for their time and feedback and are delighted that the reviewer agrees with us that it is an interesting and important finding that US AL does not outperform random sampling.
>
> ## Application to real-world graphs (addressing weaknesses 1 and 2)
>
> We follow the reviewer’s suggestion to provide more evidence for applicability of our results to real-world problems. Nonetheless, the result that US is an optimal strategy for AL still holds, even if the true epistemic uncertainty can not easily be computed in practice. We both theoretically prove and empirically (on CSBMs) this optimality. Hence, we expect a strong, disentangled UQ method to perform well in AL. Our work motivates principled UQ and shows its relationship to AL formally and does not explicitly aim at proposing a novel improved AL strategy. We updated our paper to make this point more clear.
>
> We also acknowledge the concerns regarding transferability to real-world settings. To that end, in our updated manuscript, we now introduce a method to apply our theory to real-world data in Section 6, paragraph “Real-World Data”. We detail a straightforward framework that models the epistemic uncertainty as the difference between total and aleatoric. This strategy is directly derived from our theoretic results and does not introduce any other components or tuning: The total uncertainty originates from a classifier trained solely on labeled data. We then apply the predictions from this classifier as pseudo labels to train a subsequent model on all labeled nodes (pseudo labels + real label) to model the aleatoric uncertainty. Our method outperforms many state-of-the-art US strategies on most datasets (all Figures can be seen in the updated version, Section 6 and Appendix H). We also provide a discussion of potential failure modes of this framework in Appendix H.
>
> This experiment shows that our analysis translates to real-world graphs, where we explicitly do not use any unavailable data or knowledge about the generative process. Additionally, we adapted our theory to a setting where the Bayesian classifier has to learn the parameters of the underlying unknown generative process. GNNs that predict marginal probabilities for each node fit into this framework. Since our optimality guarantees still hold under these relaxed assumptions, we believe the applicability of our findings to be even more clearly relevant to practical scenarios.
>
> We hope that with these results on real data, we can convince the reviewer that the theoretical insights on CSBMs apply to real-world graphs as the newly proposed framework is a simple straightforward implementation thereof. We want to mention that we still believe that, even though we outperform other SOTA AL strategies on multiple datasets, our theoretic analysis is the main contribution, as it bridges the gap between understanding the relationship between US and AL on graphs which has not been discussed in previous literature. As an example of how to incorporate our findings, one could try to explicitly tailor GNNs towards better modeling an assumed generative process, e.g. by considering the non-edges in a transformer-like fashion. Furthermore, as a consequence of our result that US and AL are well aligned, we strongly motivate using AL as a test of quality for newly devised uncertainty estimators (which are currently mostly evaluated on the detection of distribution shifts).
>
> We also revised our manuscript to make the structure and value of our analysis clear: We first theoretically prove the alignment between epistemic US and AL and then proceed to support this with empirical evidence. First, we focus on a simplified CSBM setting and access to all labels. This analysis isolates the effects stemming from proper disentanglement and data modeling. We then proceed to a real-world setting, where we have to rely on various approximations to apply our theoretical insights. We observe our framework to show high efficacy nonetheless. This structure enables us to both propose theoretical soundness of the results we provide as well as high practicability at the same time.
>
> ## Further changes to the manuscript
>
> **Addressing Question 1**: As we use multiple dataset splits and model initializations, all curves are already average performances. We added plots for the requested experiments including the standard deviation to the Appendix (Figure 9), such that readers can see the high variance. Further, we added a description of the repeated trial setting to the main text in Section 4.
>
> **Addressing Question 2**: We adapted the manuscript to give the background about US more concisely and added more explanation of epistemic and aleatoric effects to the background Section 2 at the same time. All additions are made visible with a blue font color in the new PDF version.

---

> > ### Author Response · Authors · 2023-11-16
> >
> > ## Final Remarks
> >
> > We again want to thank the reviewer for their time and feedback. We believe that the theoretical and empirical analysis we conducted is valuable toward the development of principled UQ on graphs. We acknowledge that the request for a practical applicable approach significantly improves our work by showcasing its relevance to real-world problems.
> >
> > We hope that the reviewer agrees with our view that our work provides a significant contribution in highlighting the alignment of the alignment of AL and US on graphs and not only providing theoretical insights but also now showing effectiveness on real-world graphs and considers raising their score.

---

> > > ### Author Response · Authors · 2023-11-21
> > >
> > > In light of the end of the author-reviewer discussion period on Wednesday, we would again like to kindly highlight our response and the manuscript's revised sections. We hope that we adequately address your concerns and are interested in your thoughts. If there are any outstanding or further questions we are delighted to discuss these.

---

> > ### Comment · Reviewer_5NFL · 2023-11-22
> >
> > Thanks for the authors' response. 1) On new real-world dataset experiments, I do see there is no advantage of your model compared with baselines; 2) In Figure 9, the variance is too high, maybe the author needs to find the reason and try to solve the unstable model performance problem.

---

> ### Author Response · Authors · 2023-11-23
>
> **A)** We are not sure why the reviewer thinks that there is no advantage over US baselines. We will emphasize that the dark green line (RHS) represents our framework in Figure 4 and Figure 15. In these figures, we show that this RHS approximation outperforms all other US strategies on all datasets! For a better quantitative comparison, we provide the area under the curve (AUC) for all of these plots which summarises the average accuracy over acquisitions in this. We highlight the best performing US strategy in bold and the best overall with a dagger.
>
>
>
> |           | Random    | Coreset | Age | ANRMAB | GEEM | Ensemble | MC-Dropout | Energy | GPN | BGCN | Aleatoric | Ours (LHS) | Ours (RHS) |
> |-----------|-----------|---------|-----|--------|------|----------|------------|--------|-----|------|-----------|------------|------------|
> |           |  |  non-US |  non-US | non-US  | non-US |         US | US | US | US | US | US | US | US |
> | CoraML    | $63.85$ | $65.23$ | $67.56$         | $61.14$ | $71.39$         | $63.47$  | $59.17$       | $63.97$ | $54.75$ | $44.45$ | $65.66$ | $67.73$ | **71.45**$^\dagger$ |
> | Citeseer  | $81.04$ | $79.38$ | $84.21$         | $81.03$ | $85.25^\dagger$ | $82.94$  | $78.86$       | $81.59$ | $65.31$ | $58.68$ | $78.85$ | $83.26$ | **83.43**         |
> | Pubmed    | $56.79$ | $64.48$ | $69.20^\dagger$ | $60.49$ | $64.82$         | $63.70$  | $58.67$       | $59.64$ | $58.82$ | $55.19$ | $61.55$ | $58.80$ | **64.36**         |
> | Photos    | $80.52$ | $82.32$ | $74.01$         | $80.92$ | $86.43^\dagger$ | $84.46$  | $72.42$       | $74.66$ | $54.78$ | $70.83$ | $71.43$ | $71.07$ | **85.52**         |
> | Computers | $72.39$ | $71.53$ | $69.31$         | $71.62$ | $74.49^\dagger$ | $68.38$  | $51.02$       | $59.62$ | $39.21$ | $58.64$ | $59.34$ | $62.16$ | **72.54**     |
>
>
>
> The only consistently equally well-performing (non-US) strategy is GEEM, which our RHS approach matches. We also point out that our method is a straightforward implementation of the theory our paper provides without tuning any models. Nonetheless, we clearly (qualitatively and quantitatively outperform other US strategies and match the best non-US strategy's performance. Thus, we are confident to conclude that our results show significant merits for practical real-world uncertainty estimation, as evidently shown by the AUC scores.
>
> **B)** We agree that the variance is extremely high on CSBMs. We point out two potential reasons: First, AL settings can inherently be noisy and since the CSBM graphs are relatively small, differences in acquired nodes (especially when picked almost randomly) may be even more pronounced. This is due to the fact that the CSMBs we study have a relatively low signal-to-noise ratio (SNR), otherwise, the whole task would be very easy to solve with GNNs (achieving near-perfect accuracy after only one acquisition per class). In this SNR regime, however, we observe high variance. Second, we study not one graph but average results over five CSBMs sampled from the same distribution: Again, due to relatively small graph sizes, we observe very different graphs in practice. We chose to average over multiple graphs to avoid “cherry-picking” one CSBM graph, where our ground-truth uncertainty performs well, and instead drew 5 random graphs to give an overall more truthful impression of performance.
>
> However, due to the new real-world graph experiments, where we show that our RHS method outperforms all US strategies, showing the superiority of ours compared to baselines on CSBMs might be obsolete. What still is interesting is the performance of our approach using various uncertainties but we can remove the other figure if the reviewer is unhappy with it.
>
> In conclusion, we want to say that we show that: (1) there is a gap between non-US AL and US AL strategies, and no SOTA method for US seems to work on par with the traditional methods. (2) we further analyze the problem and can prove that US and AL are well aligned and propose a new principled framework for US where the uncertainties are properly disentangled. (3) we show on synthetic and real-world data that we outperform all other US significantly and close the gap to non-US.
> Thus, we lay principled groundwork for new uncertainty estimators on graphs. In particular, we don’t try to ‘sell’ our method as a novel AL strategy but provide a theoretically sound and empirically well-supported analysis of the relationship of US and AL that we hope to inspire more research in this area. We firmly believe, and hope that the reviewers agree with us, that this is of value to the scientific community and in our view even worth more than proposing a new acquisition strategy.

---

### Official Review · Reviewer_fMcb · 2023-11-01

**Soundness:** 3 good
**Presentation:** 3 good
**Contribution:** 2 fair
**Rating:** 6
**Confidence:** 3

**Summary:**

This article studies the application of AL to graph data, with a focus on the approach of uncertainty sampling. The authors demonstrated through an extensive empirical analysis that many AL strategies, uncertainty-based or not, failed to surpasse random sampling on graph data. A curious observation is that uncertainty estimators which distinguish the reducible uncertainty caused by the randomness of training data from the irreducible uncertainty due to the underlying data generating process and use only the reducible uncertainty to guide the label queries work well on i.i.d. data but not on graph data. Motivated by this observation, the authors proved theoretically that, under a Contextual Stochastic Blockmodel (CSBN) with known parameters, minimizing the reducible uncertainty leads to an optimal AL strategy. This remark is  confirmed on simulated data of (CSBN).

**Strengths:**

- This work is well guided with a series of inquiries, starting with open questions in literature review, conducted with empirical observation, theoretical investigation, ending with experimental confirmation.

- The thorough empirical analysis and the original theoretical insights are of interest to the scientific community.

**Weaknesses:**

- The theoretical investigation, which is a major contribution of the article, not only considers a specific model (which is perfectly acceptable), but also assumes the full knowledge of the parameters underlying the model. As in practice the model parameters are rarely known and have to be estimated from data, their estimation error will contribute to the reducible uncertainty. Therefore defining the reducible uncertainty while assuming the model parameters to be pre-known seems to be problematic and needs at least to be discussed.

- It should be made clear earlier in the article (e.g. in the abstract or introduction) that the proposed uncertainty measure is not directly applicable in practice, and rather of theoretical interest.

**Questions:**

My questions are related to the first point of Weaknesses:

- How will the reducible uncertainty change without the knowledge of model parameters ?

- Will the conclusion regarding the optimality of using the reducible uncertainty to guide AL stay the same ?

---

> ### Author Response · Authors · 2023-11-16
>
> We want to thank the reviewer for their valuable feedback and the invested time to read our manuscript. We are delighted that the reviewer agrees with us that this is beneficial to the scientific community.
>
> ## Knowledge of model parameters (Adressing Weakness 1 and Q 1&2)
>
> We thank the reviewer for this comment and question as it is a valid point that we need to discuss in our work. We generalized our definitions to incorporate a parametrized classifier that opts to learn the true generative process:
>
> $f_\theta^*(A, X, y_O^{gt}) =  \text{argmax}_{c \in [C]^{|U|}} \textcolor{blue}{\int}\text{Pr}[{y_U= c \mid A, X, y_O = y_O^{gt}, \textcolor{blue}{\theta}}] \textcolor{blue}{p(\theta \mid A, X, y_O)d\theta}$
>
> We updated and uploaded a revision of our paper where we highlight changes in blue. The Bayesian classifier averages its prediction over its parameters $\theta$ according to a posterior distribution $p(\theta | \mathbf{X}, \mathbf{A}, \mathbf{y})$. For example, these parameters $\theta$ could be estimates of the parameters of the underlying data-generating process in the CSBM setting or GNN parameters for real data. These changes can be found in Definitions 1-3 and the corresponding proofs.
> We adapt definitions 2 and 3 accordingly. Importantly, the proposed optimality results still hold, as only the computation of the ground-truth uncertainty is affected by this change.
>
> While we can also try to learn the parameters of the underlying CSBM, this would  introduce unwarranted noise into the evaluation. We have further clarified in our manuscript that the analysis on CSBMs opts to isolate the effect of disentangling and correctly modeling uncertainty (and therefore assumes no modeling errors).
>
> In a new paragraph in Section 6, we detail an AL strategy that models the epistemic uncertainty as the difference between total and aleatoric (as in our theoretic result). Here, the total uncertainty originates from a classifier trained solely on labeled data. We then apply the predictions from this classifier as pseudo labels to train a subsequent model on all labeled nodes, with the exception of one (pseudo labels + real label) to model the aleatoric uncertainty for this one node.
>
> Our method outperforms state-of-the-art US strategies on many datasets. All Figures can be seen in the updated version, Section 6 and Appendix H).
>
> This experiment does not rely on explicit knowledge of either the generative process or unavailable labels but uses multiple approximations instead and thus could even be used as an AL strategy in practice. We, therefore, believe to provide even more valuable insights by closing the gap between an idealized synthetic setting and real scenarios. Looking ahead, future work could include improving the GNN approximators based on our other findings, such as modeling non-edges, for example in a transformer-like model.
>
> We hope that with the adaptation of our theoretic results and the new experiments we properly addressed the parameter knowledge concern. If desired by the reviewer, we are open to including an additional experiment in the CSBM setting, conducted without parameter knowledge in the camera-ready version of our work.
>
> We thank the reviewer for this pointer since framing and defining the problem in this more general parametrized way helps to better exemplify the applicability towards real-world scenarios which improves our manuscript substantially.
>
> ## Adding that we don’t propose a novel method earlier in the text
>
> We believe that the most substantial contribution of our work lies in the analysis of the problem. This groundwork paves the way for the principled development of new UQ strategies and strongly motivates AL as a valuable evaluation benchmark. For example, future studies could explore whether their uniquely disentangled uncertainties, potentially designed for different tasks, are also effective in an AL context.
>
> Therefore, while we now even showcase a straightforward and practical application of our findings, it is our intent to present it as an experimental validation of our theory's applicability to real-world graphs, rather than as the centerpiece of a methods-focused paper. Hence, we further emphasize in our updated manuscript that a new method is not the goal of our work in the introduction and hope that the reviewer agrees with our view.
>
> ## Final Comments
>
> We updated our manuscript with all the changes highlighted in blue and would be delighted to learn whether the reviewer agrees with us that these changes improve our work significantly. We again want to again thank for the reviewer's valuable feedback that helped us improve the paper and kindly ask them to consider reevaluating their score.

---

> > ### Author Response · Authors · 2023-11-21
> >
> > In light of the end of the author-reviewer discussion period on Wednesday, we would again like to kindly highlight our response and the manuscript's revised sections. We hope that we adequately address your concerns and are interested in your thoughts. If there are any outstanding or further questions we are delighted to discuss these.

---

> > ### Comment · Reviewer_fMcb · 2023-11-22
> > **Reply to the authors**
> >
> > I thank the authors for their reply. The update proposed by the authors to address my concern about the inaccessible knowledge of model parameters still seems problematic: for the aleatoric confidence, the learning of parameters should be conditioned on all the ground truth labels expect that of data point $i$, however in the equation (3) of the updated version it is conditioned on the labels of already observed instances, as is the total confidence. This is why the proposed optimality results remain unchanged.

---

> ### Author Response · Authors · 2023-11-23
>
> We want to thank the author for their answer. This is indeed a typo in our updated manuscript: instead of being conditioned on $y_O$ the parameters have to be conditioned on $y_{-I}$ when computing aleatoric confidence.  We adapted this in our document and decorated the $\theta$ with a hat to highlight that this is a different parameter belief than in Equation 2. We hope this adequately adresses the remark.
>
> The parameter learning from (partially unobserved) labels $y_{-i}$ is indeed problematic in the real world. There, one has to rely on approximations: One idea, besides our framework, would be using one of the newer graph foundation models for aleatoric learning where the model has a lot more data outside the current graph. Another idea would be to only model aleatoric uncertainty in the absence of graph effects, i.e. using a LLM on the text of e.g. CoraML.
>
> What we propose in our real-world experiment is a model that uses the pseudo labels from the learned classifier. That is, we learn $f(A, X, y_O)$ and infer all unseen data $\hat{y}_U$. Then we condition the parameter learning of an auxiliary model (for aleatoric uncertainty) on $p(\hat{\theta} | A, X, y_O, \hat{y}_U)$, i.e. we approximate $y_U$ with $\hat{y}_U$.
>
> In our study on real-world datasets, we in fact use **different parameter sets** $\theta$ and $\hat{\theta}$ for estimating total and aleatoric confidence (LHS). This fits into the framework, as we assume different parameter beliefs for both uncertainty types $p(\theta | A, X, y_O)$ and $p(\hat{\theta} | A, X, y_O, \hat{y}_U)$.
>
>
> Our results show that this method (LHS) and the other approximation we use (RHS) work well in practice: We again summarise the AUC scores for US and non-US method, where the best US strategy is highlighted bold and the best overall strategy (ie also non-US) is highlighted with a dagger in this table:
>
> |           | Random    | Coreset | Age | ANRMAB | GEEM | Ensemble | MC-Dropout | Energy | GPN | BGCN | Aleatoric | Ours (LHS) | Ours (RHS) |
> |-----------|-----------|---------|-----|--------|------|----------|------------|--------|-----|------|-----------|------------|------------|
> |           |  |  non-US |  non-US | non-US  | non-US |         US | US | US | US | US | US | US | US |
> | CoraML    | $63.85$ | $65.23$ | $67.56$         | $61.14$ | $71.39$         | $63.47$  | $59.17$       | $63.97$ | $54.75$ | $44.45$ | $65.66$ | $67.73$ | **71.45**$^\dagger$ |
> | Citeseer  | $81.04$ | $79.38$ | $84.21$         | $81.03$ | $85.25^\dagger$ | $82.94$  | $78.86$       | $81.59$ | $65.31$ | $58.68$ | $78.85$ | $83.26$ | **83.43**         |
> | Pubmed    | $56.79$ | $64.48$ | $69.20^\dagger$ | $60.49$ | $64.82$         | $63.70$  | $58.67$       | $59.64$ | $58.82$ | $55.19$ | $61.55$ | $58.80$ | **64.36**         |
> | Photos    | $80.52$ | $82.32$ | $74.01$         | $80.92$ | $86.43^\dagger$ | $84.46$  | $72.42$       | $74.66$ | $54.78$ | $70.83$ | $71.43$ | $71.07$ | **85.52**         |
> | Computers | $72.39$ | $71.53$ | $69.31$         | $71.62$ | $74.49^\dagger$ | $68.38$  | $51.02$       | $59.62$ | $39.21$ | $58.64$ | $59.34$ | $62.16$ | **72.54**     |
>
> In conclusion, we want to say that the main goal of our paper is to investigate why all SOTA US strategies fail on AL. We find that this is due to wrong modeling choices. With a simple fix, we outperform all SOTA US strategies. We hope that our findings spark even more sophisticated strategies for US on graphs, which we believe to be underdeveloped. In particular, we don’t try to ‘sell’ our method as a novel AL strategy but provide a theoretically sound and empirically well-supported analysis of the relationship of US and AL that we hope to inspire more research in this area.
> We firmly believe, and hope that the reviewer agrees with us, that this is of value to the scientific community and in our view worth more than ‘just’ proposing a new acquisition strategy.

---

### Author Response · Authors · 2023-11-16

We thank the reviewers for the effort and them pointing out that the main objective of our work was not stressed clearly enough: Even though we follow the suggestions of the reviewers and explore the applicability of our results to real-world settings, we believe that the core contribution of our work lies in both theoretically and empirically exploring, analyzing and thus motivating UQ for AL on graphs. Our contribution is threefold:

1. We provide an extensive quantitative benchmark of contemporary UQ on graphs employed in US - a facet of UQ that is completely neglected in the literature so far even though US is employed effectively for i.i.d. data.
2. We lay the theoretical groundwork for principled UQ on graphs that provably aligns well with AL and translates to realistic AL scenarios in which a Bayesian classifier learns the unknown data-generating process. **This theoretical framework is fully valid for AL on graphs in general, not only synthetic data**.
3. We provide an evaluation in an idealized setting on synthetic data that, while requiring access to unavailable ground-truth information, clearly isolates the importance of disentangling uncertainty to make US effective. We updated our manuscript to make it more clear that our analysis focuses on ground-truth uncertainty that needs to be approximated in practice and therefore can rely on unavailable information in a theoretical context. Notably, similar non-disentangled uncertainty estimators that are supplied with the same information perform worse. We also now supply an additional evaluation on real-world datasets in which no information about the true unobserved labels is leaked. We find that a straightforward approximative estimate of our proposed ground-truth uncertainty outperforms all other UQ baselines and many other AL strategies off-the-shelf. This further strengthens the empirical applicability and importance of our results to the field of developing principled UQ methods for graphs.

We explicitly refrain from framing this work as a proposal of a new AL method. Instead, we present novel theoretical insights that relate US and AL on graphs and support our findings especially both on synthetic and real data. Our study reveals important facets of modeling uncertainty on graphs in a principled way and directly points toward novel ways of approximating and disentangling uncertainty in light of our findings.

With the addition of experiments on real-world data, we believe to make a strong case for the relevance of our work not only from a theoretical perspective but also for practical applications. We explicitly want to thank the reviewers for their requests for a more practical evaluation and hope that they agree with us that the scientific community would benefit from our paper.

---

### Meta-Review · Area_Chair_tijo · 2023-12-06

**Metareview:**

The submission primarily focuses on analyzing (theoretically and empirically) the use of active learning in a graph-based setting (node classification). The investigation demonstrates the short-comings of uncertainty sampling for several existing notions of uncertainty in the graph based setting, in contrast to the standard iid learning setting. The authors go on to show that "ground-truth epistemic uncertainty" is theoretical optimal for the studied setting and performs well in a stylized empirical setting.

Reviewers agree that the insights brought by the negative results are informative and important, however, several also raised the concern that the submission lacked a practical algorithm to leverage them. During discussion the authors did provide an additional practical uncertainty estimator based approach as well as experiments, but did not fully convince reviewers.

The submission will benefit from fully fleshing out an empirical study of the practical method inspired by analysis provided in the original version of this submission, in my opinion, creating a very strong paper with a complete story.

**Justification For Why Not Higher Score:**

The initial submission provided and analysis of short comings of AL methods in the graph setting and suggested a theoretical optimal notion of uncertainty that should be used instead. Reviewers noted the lack of a practical use of the proposed notion, which the authors (admirably) responded to by providing additional experiments in a practical setting.  However, given that reviewers still do not seem to be fully convinced and, in my opinion, the addition to the paper is a rather significant change, I would suggest a rewrite and re-submission a more fully fleshed story.

**Justification For Why Not Lower Score:**

N/A

---

### Decision · Program_Chairs · 2024-01-16

Reject